# Towards Calibrated Model for Long-Tailed Visual Recognition from Prior Perspective

**Zhengzhuo Xu**[1*], **Zenghao Chai**[1*], **Chun Yuan**[1,2†]

[1]Shenzhen International Graduate School, Tsinghua University
[2]Peng Cheng Laboratory
xzz20@mails.tsinghua.edu.cn, zenghaochai@gmail.com,
yuanc@sz.tsinghua.edu.cn

## Abstract

Real-world data universally confronts a severe class-imbalance problem and exhibits a *long-tailed* distribution, i.e., most labels are associated with limited instances. The naïve models supervised by such datasets would prefer dominant labels, encounter a serious generalization challenge and become poorly calibrated. We propose two novel methods from the *prior* perspective to alleviate this dilemma. First, we deduce a balance-oriented data augmentation named Uniform Mixup (UniMix) to promote *mixup* in long-tailed scenarios, which adopts advanced mixing factor and sampler in favor of the minority. Second, motivated by the Bayesian theory, we figure out the Bayes Bias (Bayias), an inherent bias caused by the inconsistency of *prior*, and compensate it as a modification on standard cross-entropy loss. We further prove that both the proposed methods ensure the classification *calibration* theoretically and empirically. Extensive experiments verify that our strategies contribute to a better-calibrated model and their combination achieves state-of-the-art performance on CIFAR-LT, ImageNet-LT, and iNaturalist 2018.

## 1 Introduction

Balanced and large-scaled datasets [49, 39] have promoted deep neural networks to achieve remarkable success in many visual tasks [23, 48, 21]. However, real-world data typically exhibits a *long-tailed* (LT) distribution [34, 42, 25, 17], and collecting a minority category (**tail**) sample always leads to more occurrences of common classes (**head**) [53, 26], resulting in most labels associated with limited instances. The paucity of samples may cause insufficient feature learning on the tail classes [64, 12, 33, 42], and such data imbalance will bias the model towards dominant labels [52, 53, 37]. Hence, the generalization of minority categories is an enormous challenge.

The intuitive approaches such as directly over-sampling the tail [9, 4, 45, 50, 5] or under-sampling the head [18, 4, 20] will cause serious robustness problems. *mixup* [60] and its extensions [55, 59, 10] are effective feature improvement methods and contribute to a well-calibrated model in balanced datasets [54, 61], i.e., *the predicted confidence indicates actual accuracy likelihood* [16, 54]. However, *mixup* is inadequately calibrated in an imbalanced LT scenario (Fig. 1). In this paper, we raise a conception called $\xi$-*Aug* to analyze *mixup* and figure out that it tends to generate more head-head pairs, resulting in unsatisfactory generalization of the tail. Therefore, we propose Uniform Mixup (UniMix), which adopts a tail-favored *mixing factor* related to label *prior* and a *inverse sampling strategy* to encourage more head-tail pairs occurrence for better generalization and *calibration*.

---

*Equal contribution.
†Corresponding author.

35th Conference on Neural Information Processing Systems (NeurIPS 2021).

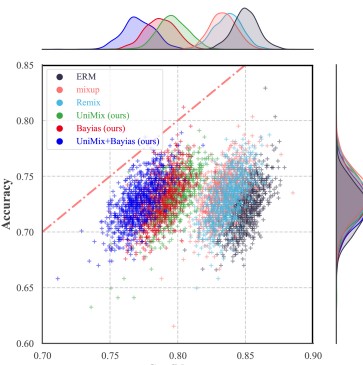

Figure 1: Joint density plots of accuracy vs. confidence to measure the *calibration* of classifiers on CIFAR-100-LT-100 during training. A well-calibrated classifier's density will lay around the red dot line $y = x$, indicating prediction score reflects the actual likelihood of accuracy. *mixup* manages to regularize classifier on balanced datasets. However, both *mixup* and its extensions tend to be overconfident in LT scenarios. Our UniMix reconstructs a more balanced dataset and Bayias-compensated CE erases *prior* bias to ensure better *calibration*. Without loss of accuracy, either of proposed methods trains the same classifier more calibrated and their combination achieves the best. How to measure *calibration* and more visualization results are available in Appendix D.2.

Previous works adjust the logits *weight* [31, 12, 52, 58] or *margin* [6, 44] on standard *Softmax* cross-entropy (CE) loss to tackle the bias towards dominant labels. We analyze the inconstancy of label *prior*, which varies in LT train set and balanced test set, and pinpoint an inherent bias named Bayes Bias (Bayias). Based on the Bayesian theory, the *posterior* is proportional to *prior* times *likelihood*. Hence, it's necessary to adjust the *posterior* on train set by compensating different *prior* for each class, which can serve as an additional *margin* on CE. We further demonstrate that the Bayias-compensated CE ensures classification *calibration* and propose a unified learning manner to combine Bayias with UniMix towards a better-calibrated model (see in Fig.1). Furthermore, we suggest that bad calibrated approaches are counterproductive with each other, which provides a heuristic way to analyze the combined results of different feature improvement and loss modification methods (see in Tab.4).

In summary, our contributions are: 1) We raise the concept of $\xi$-*Aug* to theoretically explain the reason of *mixup*'s miscalibration in LT scenarios and propose Unimix (Sec.3.1) composed of novel mixing and sampling strategies to construct a more class-balanced virtual dataset. 2) We propose the Bayias (Sec.3.2) to compensate the bias incurred by different label *prior*, which can be unified with UniMix by a training manner for better classification *calibration*. 3) We conduct sufficient experiments to demonstrate that our method trains a well-calibrated model and achieves state-of-the-art results on CIFAR-10-LT, CIFAR-100-LT, ImageNet-LT, and iNaturalist 2018.

## 2  Analysis of *mixup*

The core of supervised image classification is to find a $\theta$ parameterized mapping $\mathcal{F}_\theta : X \in \mathbb{R}^{c \times h \times w} \mapsto Y \in \mathbb{R}^{C \times 1}$ to estimate the empirical Dirac delta distribution $\mathbb{P}_\delta(x, y) = \frac{1}{N} \sum_{i=1}^{N} \delta(x_i, y_i)$ of $N$ instances $x \in \mathcal{X}$ and labels $y \in \mathcal{Y}$. The learning progress by minimizing Eq.1 is known as Empirical Risk Minimization (ERM), where $\mathcal{L}(Y = y_i, \mathcal{F}_\theta(X = x_i))$ is $x_i$'s conditional risk.

$$R_\delta(\mathcal{F}_\theta) = \int_{x \in \mathcal{X}} \mathcal{L}\left(Y = y, \mathcal{F}_\theta(X = x)\right) d\mathbb{P}_\delta(x, y) = \frac{1}{N} \sum_{i=1}^{N} \mathcal{L}\left(Y = y_i, \mathcal{F}_\theta(X = x_i)\right) \quad (1)$$

To overcome the over-fitting caused by insufficient training of $N$ samples, *mixup* utilizes Eq.2 to extend the feature space to its vicinity based on Vicinal Risk Minimization (VRM) [8].

$$\widetilde{x} = \xi \cdot x_i + (1 - \xi) \cdot x_j \qquad \widetilde{y} = \xi \cdot y_i + (1 - \xi) \cdot y_j \quad (2)$$

where $\xi \sim Beta(\alpha, \alpha), \alpha \in [0, 1]$, the sample pair $(x_i, y_i), (x_j, y_j)$ is drawn from training dataset $\mathcal{D}_{train}$ randomly. Hence, Eq.2 converts $\mathbb{P}_\delta(X, Y)$ into empirical *vicinal distribution* $\mathbb{P}_\nu(\widetilde{x}, \widetilde{y}) = \frac{1}{N} \sum_{i=1}^{N} \nu(\widetilde{x}, \widetilde{y}|x_i, y_i)$, where $\nu(\cdot)$ describes the manner of finding virtual pairs $(\widetilde{x}, \widetilde{y})$ in the vicinity of arbitrary sample $(x_i, y_i)$. Then, we construct a new dataset $\mathcal{D}_\nu := \{(\widetilde{x}_k, \widetilde{y}_k)\}_{k=1}^{M}$ via Eq.2 and minimize the empirical vicinal risk by Vicinal Risk Minimization (VRM):

$$R_\nu(\mathcal{F}_\theta) = \int_{\widetilde{x} \in \widetilde{\mathcal{X}}} \mathcal{L}\left(\widetilde{Y} = \widetilde{y}, \mathcal{F}_\theta(\widetilde{X} = \widetilde{x})\right) d\mathbb{P}_\nu(\widetilde{x}, \widetilde{y}) = \frac{1}{M} \sum_{i=1}^{M} \mathcal{L}\left(\widetilde{Y} = \widetilde{y}_i, \mathcal{F}_\theta(\widetilde{X} = \widetilde{x}_i)\right) \quad (3)$$

*mixup* is proven to be effective on balanced dataset due to its improvement of *calibration* [54, 16], but it is unsatisfactory in LT scenarios (see in Tab.4). In Fig.1, *mixup* fails to train a calibrated model,

which surpasses baseline (ERM) a little in accuracy and seldom contributes to *calibration* (far from $y = x$). To analyze the insufficiency of *mixup*, the definition of $\xi$-*Aug* is raised.

**Definition 1** $\xi$-*Aug.* *The virtual sample* $(\widetilde{x}_{i,j}, \widetilde{y}_{i,j})$ *generated by Eq.2 with mixing factor* $\xi$ *is defined as a* $\xi$-*Aug sample, which is a robust sample of class* $y_i$ *(class* $y_j$*) iff* $\xi \geq 0.5(\xi < 0.5)$ *that contributes to class* $y_i$ *(class* $y_j$*) in model's feature learning.*

In LT scenarios, we reasonably assume the instance number $n$ of each class is *exponential* with parameter $\lambda$ [12] if indices are descending sorted by $n_{y_i}$, where $y_i \in [1, C]$ and $C$ is the total class number. Generally, the imbalance factor is defined as $\rho = n_{y_1}/n_{y_C}$ to measure how skewed the LT dataset is. It is easy to draw $\lambda = \ln \rho/(C - 1)$. Hence, we can describe the LT dataset as Eq.4:

$$\mathbb{P}(Y = y_i) = \frac{\iint_{x_i \in \mathcal{X}, y_j \in \mathcal{Y}} \mathbb{1}(X = x_i, Y = y_i)dx_i dy_j}{\iint_{x_i \in \mathcal{X}, y_j \in \mathcal{Y}} \mathbb{1}(X = x_i, Y = y_j)dx_i dy_j} = \frac{\lambda}{e^{-\lambda} - e^{-\lambda C}} e^{-\lambda y_i}, y_i \in [1, C] \quad (4)$$

Then, we derive the following corollary to illustrate the limitation of naïve *mixup* strategy.

**Corollary 1** *When* $\boldsymbol{\xi \sim Beta(\alpha, \alpha)}, \alpha \in [0, 1]$, *the newly mixed dataset* $\mathcal{D}_\nu$ *composed of* $\xi$-*Aug samples* $(\widetilde{x}_{i,j}, \widetilde{y}_{i,j})$ *follows the same long-tailed distribution as the origin dataset* $\mathcal{D}_{train}$, *where* $(x_i, y_i)$ *and* $(x_j, y_j)$ *are **randomly** sampled from* $\mathcal{D}_{train}$. *(See detail derivation in Appendix A.2.)*

$$\mathbb{P}_{mixup}(Y^* = y_i) = \mathbb{P}^2(Y = y_i) + \mathbb{P}(Y = y_i) \iint_{y_i \neq y_j} Beta(\alpha, \alpha)\mathbb{P}(Y = y_j)d\xi dy_j$$
$$= \frac{\lambda}{e^{-\lambda} - e^{-\lambda C}} e^{-\lambda y_i}, y_i \in [1, C] \quad (5)$$

In *mixup*, the probability of any $(\widetilde{x}_{i,j}, \widetilde{y}_{i,j})$ belongs to class $y_i$ or class $y_j$ is strictly determined by $\xi$ and $\mathbb{E}(\xi) \equiv 0.5$. Furthermore, both $(x_i, y_i)$ and $(x_j, y_j)$ are randomly sampled and concentrated on the head instead of tail, resulting in that the head classes get more $\xi$-*Aug* samples than the tail ones.

## 3 Methodology

### 3.1 UniMix: balance-oriented feature improvement

*mixup* and its extensions tend to generate head-majority pseudo data, which leads to the deficiency on the tail feature learning and results in a bad-calibrated model. To obtain a more balanced dataset $\mathcal{D}_\nu$, we propose the UniMix Factor $\xi^*_{i,j}$ related to the *prior* probability of each category and a novel UniMix Sampler to obtain sample pairs. Our motivation is to generate comparable $\xi$-*Aug* samples of each class for better generalization and *calibration*.

**UniMix Factor.** Specifically, the *prior* in imbalanced train set and balanced test set of class $y_i$ is defined as $\mathbb{P}_{train}(Y = y_i) \triangleq \pi_{y_i}$, and $\mathbb{P}_{test}(Y = y_i) \equiv 1/C$, respectively. We design the UniMix Factor $\xi^*_{i,j}$ for each virtual sample $\widetilde{x}_{i,j}$ instead of a fixed $\xi$ in *mixup*. Consider adjusting $\xi$ with the class *prior* probability $\pi_{y_i}, \pi_{y_j}$. It is intuitive that a proper factor $\xi_{i,j} = \pi_{y_j}/(\pi_{y_i} + \pi_{y_j})$ ensures $\widetilde{x}_{i,j}$ to be a $\xi$-*Aug* sample of class $y_j$ if $\pi_{y_i} \geq \pi_{y_j}$, i.e., class $y_i$ occupies more instances than class $y_j$.

However, $\xi_{i,j}$ is uniquely determined by $\pi_{y_i}, \pi_{y_j}$. To improve the robustness and generalization, original $Beta(\alpha, \alpha)$ is adjusted to obtain UniMix Factor $\xi^*_{i,j}$. Notice that $\xi$ is close to 0 or 1 and symmetric at 0.5, we transform it to maximize the probability of $\xi_{i,j} = \pi_{y_j}/(\pi_{y_i} + \pi_{y_j})$ and its vicinity. Specifically, if note $\xi \sim Beta(\alpha, \alpha)$ as $f(\xi; \alpha, \alpha)$, we define $\xi^*_{i,j} \sim \mathscr{U}(\pi_{y_i}, \pi_{y_j}, \alpha, \alpha)$ as:

$$\xi^*_{i,j} \sim \mathscr{U}(\pi_{y_i}, \pi_{y_j}, \alpha, \alpha) = \begin{cases} f(\xi^*_{i,j} - \dfrac{\pi_{y_j}}{\pi_{y_i} + \pi_{y_j}} + 1; \alpha, \alpha), & \xi^*_{i,j} \in [0, \dfrac{\pi_{y_j}}{\pi_{y_i} + \pi_{y_j}}); \\ f(\xi^*_{i,j} - \dfrac{\pi_{y_j}}{\pi_{y_i} + \pi_{y_j}}; \alpha, \alpha), & \xi^*_{i,j} \in [\dfrac{\pi_{y_j}}{\pi_{y_i} + \pi_{y_j}}, 1] \end{cases} \quad (6)$$

Rethink Eq.2 with $\xi^*_{i,j}$ described as Eq.6:

$$\widetilde{x}_{i,j} = \xi^*_{i,j} \cdot x_i + (1 - \xi^*_{i,j}) \cdot x_j \qquad \widetilde{y}_{i,j} = \xi^*_{i,j} \cdot y_i + (1 - \xi^*_{i,j}) \cdot y_j \quad (7)$$

We have the following corollary to show how $\xi^*_{i,j}$ ameliorates the imbalance of $\mathcal{D}_{train}$:

**Corollary 2** *When $\xi_{i,j}^* \sim \mathscr{U}(\pi_{y_i}, \pi_{y_j}, \alpha, \alpha), \alpha \in [0,1]$, the newly mixed dataset $\mathcal{D}_\nu$ composed of $\xi$-Aug samples $(\widetilde{x}_{i,j}, \widetilde{y}_{i,j})$ follows a middle-majority distribution (see Fig.2), where $(x_i, y_i)$ and $(x_j, y_j)$ are both **randomly** sampled from $\mathcal{D}_{train}$. (See detail derivation in Appendix A.3.)*

$$
\mathbb{P}^*_{mixup}(Y^* = y_i) = \mathbb{P}(Y = y_i) \int_{y_j < y_i} \mathbb{1}\left( \int \xi_{i,j}^* \mathscr{U}(\pi_i, \pi_j, \alpha, \alpha) d\xi_{i,j}^* \geq 0.5 \right) \mathbb{P}(Y = y_j) dy_j
$$
$$
= \frac{\lambda}{(e^{-\lambda} - e^{-\lambda C})^2} \left( e^{-\lambda(y_i+1)} - e^{-2\lambda y_i} \right), y_i \in [1, C] \tag{8}
$$

**UniMix Sampler.** UniMix Factor facilitates $\xi$-*Aug* samples more balance-distributed over all classes. However, most samples are still $\xi$-*Aug* for the head or middle (see Fig.2(green)). Actually, the constraint that pair $x_i, x_j$ drawn from the head and tail respectively is preferred, which dominantly generates $\xi$-*Aug* samples for tail classes with $\xi_{i,j}^*$. To this end, we consider sample $x_j$ from $\mathcal{D}_{train}$ with probability inverse to the label *prior*:

$$
\mathbb{P}_{inv}(Y = y_i) = \frac{\mathbb{P}^\tau(Y = y_i)}{\int_{y_j \in \mathcal{Y}} \mathbb{P}^\tau(Y = y_j) dy_j} \tag{9}
$$

When $\tau = 1$, UniMix Sampler is equivalent to a random sampler. $\tau < 1$ indicates that $x_j$ has higher probability drawn from tail class. Note that $x_i$ is still randomly sampled from $\mathcal{D}_{train}$, i.e., it's most likely drawn from the majority class. The virtual sample $\widetilde{x}_{i,j}$ obtained in this manner is mainly a $\xi$-*Aug* sample of the tail composite with $x_i$ from the head. Hence Corollary 3 is derived:

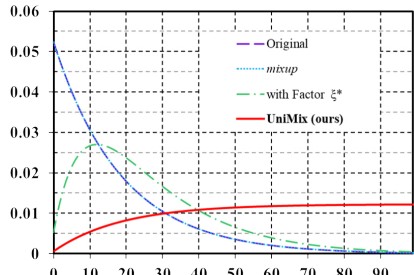

Figure 2: Visualization of $\xi$-*Aug* samples distribution ($C = 100, \rho = 200$) in Corollary 1,2,3. *x*-axis: class indices. *y*-axis: probability of each class. *mixup* (blue) exhibits the same LT distribution as origin (purple). $\xi^*$ (green) alleviates such situation and the full pipeline ($\tau = -1$) (red) constructs a more uniform distributed dataset. See more results in Appendix A.5.

**Corollary 3** *When $\xi_{i,j}^* \sim \mathscr{U}(\pi_{y_i}, \pi_{y_j}, \alpha, \alpha), \alpha \in [0,1]$, the newly mixed dataset $\mathcal{D}_\nu$ composed of $\xi$-Aug samples $(\widetilde{x}_{i,j}, \widetilde{y}_{i,j})$ follows a tail-majority distribution (see Fig.2), where $(x_i, y_i)$ is **randomly** and $(x_j, y_j)$ is **inversely** sampled from $\mathcal{D}_{train}$, respectively. (See detail derivation in Appendix A.4.)*

$$
\mathbb{P}_{UniMix}(Y^* = y_i) = \mathbb{P}(Y = y_i) \int_{y_j < y_i} \mathbb{1}\left( \int \xi_{i,j}^* \mathscr{U}(\pi_i, \pi_j, \alpha, \alpha) d\xi_{i,j}^* \geq 0.5 \right) \mathbb{P}_{inv}(Y = y_j) dy_j
$$
$$
= \frac{\lambda}{(e^{-\lambda} - e^{-\lambda C})(e^{-\lambda \tau C} - e^{-\lambda \tau})} \left( e^{-\lambda y_i(\tau+1)} - e^{-\lambda(\tau+y_i)} \right), y_i \in [1, C] \tag{10}
$$

With the proposed UniMix Factor and UniMix Sampler, we get the complete UniMix manner, which constructs a uniform $\xi$-*Aug* samples distribution for VRM and greatly facilitates model's *calibration* (See Fig.2 (red) & 1). We construct $\mathcal{D}_\nu := \{(\widetilde{x}_k, \widetilde{y}_k)\}_{k=1}^M$ where $\{\widetilde{x}_k, \widetilde{y}_k\}$ is $(\widetilde{x}_{i,j}, \widetilde{y}_{i,j})$ generated by $(x_i, y_i)$ and $(x_j, y_j)$. We conduct training via Eq.3 and the loss via VRM is available as:

$$
\mathcal{L}(\widetilde{y}_k, \mathcal{F}_\theta(\widetilde{x}_k)) = \xi_{i,j}^* \mathcal{L}(y_i, \mathcal{F}_\theta(\widetilde{x}_{i,j})) + (1 - \xi_{i,j}^*) \mathcal{L}(y_j, \mathcal{F}_\theta(\widetilde{x}_{i,j})) \tag{11}
$$

### 3.2 Bayias: an inherent bias in LT

The bias between LT set and balanced set is ineluctable and numerous studies [12, 53, 57] have demonstrated its existence. To eliminate the systematic bias that classifier tends to predict the head, we reconsider the parameters training process. Generally, a classifier can be modeled as:

$$
\hat{y} = \arg\max_{y_i \in \mathcal{Y}} \frac{e^{\sum_{d_i \in D}[(W^T)_{y_i}^{(d_i)} \mathcal{F}(x;\theta)^{(d_i)}] + b_{y_i}}}{\sum_{y_j \in \mathcal{Y}} e^{\sum_{d_i \in D}[(W^T)_{y_j}^{(d_i)} \mathcal{F}(x;\theta)^{(d_i)}] + b_{y_j}}} \triangleq \arg\max_{y_i \in \mathcal{Y}} \frac{e^{\psi(x;\theta,W,b)_{y_i}}}{\sum_{y_j \in \mathcal{Y}} e^{\psi(x;\theta,W,b)_{y_j}}} \tag{12}
$$

where $\hat{y}$ indicts the predicted label, and $\mathcal{F}(x;\theta) \in \mathbb{R}^{D \times 1}$ is the $D$-dimension feature extracted by the backbone with parameter $\theta$. $W \in \mathbb{R}^{D \times C}$ represents the parameter matrix of the classifier.

Previous works [12, 53] have demonstrated that it is not suitable for imbalance learning if one ignores such bias. In LT scenarios, the instances number in each class of the train set varies greatly, which means the corresponding *prior* probability $\mathbb{P}_{train}(Y = y)$ is highly skewed whereas the distribution on the test set $\mathbb{P}_{test}(Y = y)$ is uniform.

According to Bayesian theory, *posterior* is proportional to *prior* times *likelihood*. The supervised training process of $\psi(x; \theta, W, b)$ in Eq.12 can regard as the estimation of *likelihood*, which is equivalent to get *posterior* for inference in balanced dataset. Considering the difference of *prior* during training and testing, we have the following theorem (See detail derivation in Appendix B.1):

**Theorem 3.1** *For classification, let $\psi(x; \theta, W, b)$ be a hypothesis class of neural networks of input $X = x$, the classification with Softmax should contain the influence of prior, i.e., the predicted label during training should be:*

$$\hat{y} = \arg\max_{y_i \in \mathcal{Y}} \frac{e^{\psi(x;\theta,W,b)_{y_i} + \log(\boldsymbol{\pi_{y_i}}) + \log(C)}}{\sum_{y_j \in \mathcal{Y}} e^{\psi(x;\theta,W,b)_{y_j} + \log(\boldsymbol{\pi_{y_j}}) + \log(C)}} \tag{13}$$

In balanced datasets, all classes share the same *prior*. Hence, the supervised model $\psi(x; \theta, W, b)$ could use the estimated *likelihood* $\mathbb{P}(X = x|Y = y)$ of train set to correctly obtain *posterior* $\mathbb{P}(Y = y|X = x)$ in test set. However, in LT datasets where $\mathbb{P}_{train}(Y = y_i) = \pi_{y_i}$ and $\mathbb{P}_{test}(Y = y_i) \equiv 1/C$, *prior* cannot be regard as a constant over all classes any more. Due to the difference on *prior*, the learned parameters $\theta, W, b \triangleq \Theta$ will yield class-level bias, i.e., the optimization direction is no longer as described in Eq.12. Thus, the bias incurred by *prior* should compensate at first. To correctness the bias for inferring, the offset term that model in LT dataset to compensate is:

$$\mathscr{B}_y = \log(\pi_y) + \log(C) \tag{14}$$

Furthermore, the proposed Bayias $\mathscr{B}_y$ enables predicted probability reflecting the actual correctness likelihood, expressed as Theorem 3.2. (See detail derivation in Appendix B.2.)

**Theorem 3.2** $\mathscr{B}_y$-*compensated cross-entropy loss in Eq.15 ensures classification calibration.*

$$\mathcal{L}_{\mathscr{B}}(y_i, \psi(x; \Theta)) = \log\left[1 + \sum_{y_k \neq y_i} e^{(\mathscr{B}_{y_k} - \mathscr{B}_{y_i})} \cdot e^{\psi(x;\Theta)_{y_k} - \psi(x;\Theta)_{y_i}}\right] \tag{15}$$

Here, the optimization direction during training will convert to $\psi(X; \theta, W, b) + \mathscr{B}_y$. In particular, if the train set is balanced, $\mathbb{P}_{train}(Y = y) \triangleq \pi_y \equiv 1/C$, then $\mathscr{B}_y = \log(1/C) + \log(C) \equiv 0$, which means the Eq.12 is a balanced case of Eq.13. We further raise that $\mathscr{B}_y$ is critical to the classification *calibration* in Theorem 3.2. The pairwise loss in Eq.15 will guide model to avoid over-fitting the tail or under-fitting the head with better generalization, which contributes to a better calibrated model.

Compared with logit adjustment [44], which is also a *margin* modification, it is necessary to make a clear statement about the concrete difference from three points. 1) Logit adjustment is motivated by Balanced Error Rate (BER), while the Bayias compensated CE loss is inspired by the Bayesian theorem. We focus more on the model performance on the real-world data distribution. 2) As motioned above, our loss is consistent with standard CE loss when the train set label prior is the same as real test label distribution. 3) Our loss can tackle the imbalanced test set situation as well by simply setting the margin as $\mathscr{B}_y = \log(\pi_y) + \log(\pi'_y)$, where the $\pi'_y$ represents the test label distribution. The experiment evidence can be found in Appendix Tab.D2.

### 3.3 Towards calibrated model with UniMix and Bayias

It's intuitive to integrate feature improvement methods with loss modification ones for better performance. However, we find such combinations fail in most cases and are counterproductive with each other, i.e., the combined methods reach unsatisfactory performance gains. We suspect that these methods take contradictory trade-offs and thus result in overconfidence and bad *calibration*. Fortunately, the proposed UniMix and Bayias are both proven to ensure *calibration*. To achieve a better-calibrated model for superior performance gains, we introduce Alg.1 to tackle the previous dilemma and integrate our two proposed approaches to deal with poor generalization of tail classes. Specially, inspired by previous work [24], we overcome the coverage difficulty in *mixup* [60] by removing UniMix in the last several epochs and thus maintain the same epoch as baselines. Note that Bayias-compensated CE is only adopted in the training process as discussed in Sec.3.2.

---

**Algorithm 1** Integrated training manner towards calibrated model.

---

**Input:** $\mathcal{D}_{train}$, Batch Size $\mathcal{N}$, Stop Steps $T_1, T_2$, Random Sampler $\mathcal{R}$, UniMix Sampler $\mathcal{R}^*$
**Output:** Optimized $\Theta^*$, i.e., feature extractor parameters $\theta^*$, classifier parameters $W^*, b^*$

1: Initialize the parameters $\Theta^{(0)}$ randomly and calculate $\mathscr{B}_y$ via Eq.14
2: **for** $t = 0$ to $T_1$ **do**
3:      Sample a mini-batch $\mathcal{B} = \{x_i, y_i\}_{i=1}^{\mathcal{N}} \leftarrow \mathcal{R}(\mathcal{D}_{train}, \mathcal{N})$
4:      Sample a mini-batch $\mathcal{B}^* = \{x_j^*, y_j^*\}_{j=1}^{\mathcal{N}} \leftarrow \mathcal{R}^*(\mathcal{D}_{train}, \mathcal{N})$
5:      Calculate UniMix factor $\xi^*$ via Eq.6
6:      Construct VRM dataset $\mathcal{B}_\nu = \{\widetilde{x}_k, \widetilde{y}_k\}_{k=1}^{\mathcal{N}}$ via Eq.7
7:      Calculate $\mathcal{L}_{\mathcal{B}_\nu} = \mathbb{E}[\xi_{i,j}^* \mathcal{L}_{\mathscr{B}}(y_i, \psi(\widetilde{x}; \Theta^{(t)})) + (1 - \xi_{i,j}^*)\mathcal{L}_{\mathscr{B}}(y_j^*, \psi(\widetilde{x}; \Theta^{(t)}))]$ via Eq.11,15
8:      Update $\Theta^{(t+1)} \leftarrow \Theta^{(t)} - \alpha\nabla_{\Theta^{(t)}}\mathcal{L}_{\mathcal{B}_\nu}$
9: **end for**
10: **for** $t = T_1$ to $T_2$ **do**
11:      Sample a mini-batch $\mathcal{B} = \{x_i, y_i\}_{i=1}^{\mathcal{N}} \leftarrow \mathcal{R}(\mathcal{D}_{train}, \mathcal{N})$
12:      Calculate $\mathcal{L}_{\mathcal{B}} = \mathbb{E}[\mathcal{L}_{\mathscr{B}}(y_i, \psi(x_i; \Theta))]$ via Eq.15
13:      Update $\Theta^{(t+1)} \leftarrow \Theta^{(t)} - \alpha\nabla_{\Theta^{(t)}}\mathcal{L}_{\mathcal{B}}$
14: **end for**

---

# 4 Experiment

## 4.1 Results on synthetic dataset

We make an ideal binary classification using Support Vector Machine (SVM) [14] to show the distinguish effectiveness of UniMix. Suppose there are samples from two disjoint circles respectively:

$$z^+ = \{(x, y)|(x - x_0)^2 + (y - y_0)^2 \leq r^2\}$$
$$z^- = \{(x, y)|(x + x_0)^2 + (y + y_0)^2 \leq r^2\} \tag{16}$$

To this end, we randomly sample $m$ discrete point pairs from $z^+$ to compose positive samples $z_p^+ = \{(x_1^+, y_1^+), \cdots, (x_m^+, y_m^+)\}$, and $m$ negative samples $z_n^- = \{(x_1^-, y_1^+), \cdots, (x_m^-, y_m^-)\}$ from $z^-$ correspondingly, thus to generate a balanced dataset $\mathcal{D}_{bal} = \{z_p^+, z_n^-\}$ with $\mathbb{P}\left((x,y) \in z_p^+\right) = \mathbb{P}\left((x,y) \in z_n^-\right) = 0.5$. For imbalance data, we sample $n(n \ll m)$ negative data from $z^-$ to generate $z_{n'}^- = \{(x_1'^-, y_1'^-), \cdots, (x_n'^-, y_n'^-)\}$, so as to compose the imbalance dataset $\mathcal{D}_{imbal} = \{z_p^+, z_{n'}^-\}$, with $\mathbb{P}\left((x,y) \in z_p^+\right) \gg \mathbb{P}\left((x,y) \in z_{n'}^-\right)$. We train the SVM model on the two synthetic datasets, and visualize the classification boundary of each dataset in Fig.3.

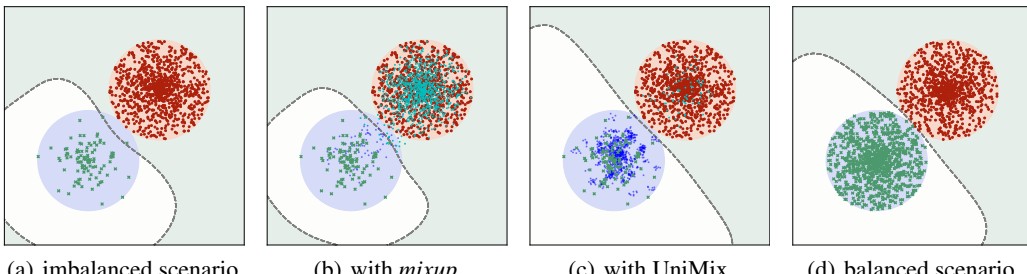

    (a) imbalanced scenario      (b) with *mixup*      (c) with UniMix      (d) balanced scenario

Figure 3: SVM decision boundary on the synthetic balanced dataset (Fig.3(d)) and imbalanced dataset (Fig.3(a),3(b),3(c)). The theoretical classification boundary of the synthetic dataset is $y=-x$. "∘" represents generated pseudo data, where blue and green represent belong to $z^-$ and $z^+$, respectively.

The SVM reaches an approximate ideal boundary on balanced datasets (Fig.3(d)) but severely deviates from the $y = -x$ in the imbalanced dataset (Fig.3(a)). As proven in Sec.2, *mixup* (Fig.3(b)) is incapable of shifting imbalance distribution, resulting in no better result than the original one (Fig.3(a)). After adopting the proposed UniMix, the classification boundary in Fig.3(c) shows much better results than the original imbalanced dataset, which gets closed to the ideal boundary.

## 4.2 Results on CIFAR-LT

The imbalanced datasets CIFAR-10-LT and CIFAR-100-LT are constructed via suitably discarding training samples following previous works [64, 12, 44, 6]. The instance numbers exponentially decay per class in train dataset and keep balanced during inference. We extensively adopt $\rho \in \{10, 50, 100, 200\}$ for comprehensive comparisons. See implementation details in Appendix C.1.

**Comparison methods.** We evaluate the proposed method against various representative and effective approaches extensively, summarized into the following groups: **a) Baseline.** We conduct plain training with CE loss called ERM as baseline. **b) Feature improvement methods** modify the input feature to cope with LT datasets. *mixup* [60] convexly combines images and labels to build virtual data for VRM. Manifold mixup [55] performs the linear combination in latent states. Remix [10] conducts the same combination on images and adopts tail-favored rules on labels. M2m [33] converts majority images to minority ones by adding noise perturbation, which need an additional pre-trained classifier. BBN [64] utilizes features from two branches in a cumulative learning manner. **c) Loss modification methods** either adjust the logits *weight* or *margin* before the *Softmax* operation. Specifically, focal loss [38], CB [12] and CDT [58] re-weight the logits with elaborate strategies, while LDAM [6] and Logit Adjustment [44] add the logits *margin* to shift decision boundary away from tail classes. **d) Other methods.** We also compare the proposed method with other two-stage approaches (e.g. DRW [6]) for comprehensive comparisons.

Table 1: Top-1 validation accuracy(%) of ResNet-32 on CIFAR-10/100-LT. E2E: end to end training. Underscore: the best performance in each group. †: our reproduced results. ‡: reported results in [64]. ⋆: reported results in [58]. Our *calibration* ensured method achieves the best performance.

| Dataset | E2E | CIFAR-10-LT | | | | CIFAR-100-LT | | | |
|---|---|---|---|---|---|---|---|---|---|
| $\rho$ (easy → hard) | - | 10 | 50 | 100 | 200 | 10 | 50 | 100 | 200 |
| ERM† | ✓ | 86.39 | 74.94 | 70.36 | 66.21 | 55.70 | 44.02 | 38.32 | 34.56 |
| *mixup*‡ [60] | ✓ | 87.10 | 77.82 | 73.06 | 67.73 | 58.02 | 44.99 | 39.54 | 34.97 |
| Manifold Mixup‡ [55] | ✓ | 87.03 | 77.95 | 72.96 | - | 56.55 | 43.09 | 38.25 | - |
| Remix [10] | ✓ | 88.15 | 79.20 | 75.36 | 67.08 | 59.36 | 46.21 | 41.94 | 36.99 |
| M2m [33] | ✗ | 87.90 | - | 78.30 | - | 58.20 | - | 42.90 | - |
| BBN‡ [64] | ✗ | 88.32 | 82.18 | 79.82 | - | 59.12 | 47.02 | 42.56 | - |
| Focal⋆ [38] | ✓ | 86.55 | 76.71† | 70.43 | 65.85 | 55.78 | 44.32† | 38.41 | 35.62 |
| Urtasun et al [47] | ✓ | 82.12 | 76.45 | 72.23 | 66.25 | 52.12 | 43.17 | 38.90 | 33.00 |
| CB-Focal [12] | ✓ | 87.10 | 79.22 | 74.57 | 68.15 | 57.99 | 45.21 | 39.60 | 36.23 |
| $\tau$-norm⋆ [30] | ✓ | 87.80 | 82.78† | 75.10 | 70.30 | 59.10 | 48.23† | 43.60 | 39.30 |
| LDAM† [6] | ✓ | 86.96 | 79.84 | 74.47 | 69.50 | 56.91 | 46.16 | 41.76 | 37.73 |
| LDAM+DRW† [6] | ✗ | 88.16 | 81.27 | 77.03 | 74.74 | 58.71 | 47.97 | 42.04 | 38.45 |
| CDT⋆ [58] | ✓ | 89.40 | 81.97† | 79.40 | 74.70 | 58.90 | 45.15† | 44.30 | 40.50 |
| Logit Adjustment [44] | ✓ | 89.26† | 83.38† | 79.91 | 75.13† | 59.87† | 49.76† | 43.89 | 40.87† |
| Ours | ✓ | 89.66 | 84.32 | 82.75 | 78.48 | 61.25 | 51.11 | 45.45 | 42.07 |

**Results.** We present results of CIFAR-10-LT and CIFAR-100-LT in Tab. 1. Our proposed method achieves state-of-the-art results against others on each $\rho$, with performance gains improved as $\rho$ gets increased (See Appendix D.1). Specifically, our method overcomes the ignorance in tail classes effectively with better *calibration*, which integrates advantages of two group approaches and thus surpass most two-stage methods (i.e., BBN, M2m, LDAM+DRW). However, not all combinations can get ideal performance gains as expected. More details will be discussed in Sec. 4.4.

To quantitatively describe the contribution to model *calibration*, we make quantitative comparisons on CIFAR-10-LT and CIFAR-100-LT. According to the definition ECE and MCE (see Appendix Eq. D.2 D.3), a well-calibrated model should minimize the ECE and MCE for better generalization and robustness. In this experiment, we adopt the most representative $\rho \in \{10, 100\}$ with previous mainstream state-of-the-art methods.

The results in Tab. 2 show that either of the proposed methods generally outperforms previous methods, and their combination enables better classification *calibration* with smaller ECE and MCE. Specifically, *mixup* and Remix have negligible contributions to model *calibration*. As analyzed before, such methods tend to generate head-head pairs in favor of the feature learning of majority

Table 2: Quantitative *calibration* metric of ResNet-32 on CIFAR-10/100-LT test set. Smaller ECE and MCE indicate better *calibration* results. Either of the proposed methods achieves a well-calibrated model compared with others. The combination of UniMix and Bayias achieves the best performance.

| Dataset | CIFAR-10-LT | | | | CIFAR-100-LT | | | |
|---|---|---|---|---|---|---|---|---|
| $\rho$ | 10 | | 100 | | 10 | | 100 | |
| *Calibration* Metric (%) | ECE | MCE | ECE | MCE | ECE | MCE | ECE | MCE |
| ERM | 6.60 | 74.96 | 20.53 | 73.91 | 22.85 | 34.50 | 38.23 | 87.22 |
| *mixup* [60] | 6.55 | 24.54 | 19.20 | 37.84 | 19.69 | 38.53 | 32.72 | 50.46 |
| Remix [10] | 6.81 | 22.44 | 15.38 | 27.99 | 20.17 | 32.99 | 33.56 | 50.96 |
| LDAM+DRW [6] | 11.22 | 45.92 | 19.89 | 49.07 | 30.54 | 55.57 | 42.18 | 64.78 |
| UniMix (ours) | 6.00 | 25.99 | 12.87 | 28.30 | 19.38 | 33.40 | 27.12 | 41.46 |
| Bayias (ours) | 5.52 | 20.14 | 11.05 | **23.72** | 17.42 | 28.26 | 24.31 | 39.66 |
| UniMix+Bayias (ours) | **4.74** | **13.67** | **10.19** | 25.47 | **15.24** | **23.67** | **23.04** | **37.36** |

classes. However, more head-tail pairs are required for better feature representation of the tail classes. In contrast, both the proposed UniMix and Bayias pay more attention to the tail and achieve satisfactory results. It is worth mentioning that improving *calibration* in post-hoc manners [16, 63] is also effective, and we will discuss it in Appendix D.2.3. Note that LDAM is even worse calibrated compared with baseline. We suggest that LDAM adopts an additional margin only for the ground-truth label from the angular perspective, which shifts the decision boundary away from the tail class and makes the tail predicting score tend to be larger. Additionally, LDAM requires the normalization of input features and classifier weight matrix. Although a scale factor is proposed to enlarge the logits for better *Softmax* operation [56], it is still harmful to *calibration*. It also accounts for its contradiction with other methods. Miscalibration methods combined will make models become even more overconfident and damage the generalization and robustness severely.

## 4.3 Results on large-scale datasets

We further verify the proposed method's effectiveness quantitatively on large-scale imbalanced datasets, i.e. ImageNet-LT and iNaturalist 2018. ImageNet-LT is the LT version of ImageNet [49] by sampling a subset following *Pareto* distribution, which contains about $115K$ images from $1,000$ classes. The number of images per class varies from 5 to $1,280$ exponentially, i.e., $\rho = 256$. In our experiment, we utilize the balanced validation set constructed by Cui *et al.* [12] for fair comparisons. The iNaturalist species classification dataset [25] is a large-scale real-world dataset which suffers from extremely label LT distribution and fine-grained problems [25, 64]. It is composed of $435,713$ images over $8,142$ classes with $\rho = 500$. The official splits of train and validation images [6, 64, 30] are adopted for fair comparisons. See implementation details in Appendix C.2.

Table 3: Top-1 validation accuracy(%) of ResNet-10/50 on ImageNet-LT and ResNet-50 on iNaturalist 2018. E2E: end to end training. †: our reproduced results. ‡: results reported in origin paper.

| Dataset | | ImageNet-LT | | | | iNaturalist 2018 | |
|---|---|---|---|---|---|---|---|
| Method | E2E | ResNet-10 | $\Delta$ | ResNet-50 | $\Delta$ | ResNet-50 | $\Delta$ |
| CE† | ✓ | 35.88 | - | 38.88 | - | 60.88 | - |
| CB-CE† [12] | ✓ | 37.06 | +1.18 | 40.85 | +1.97 | 63.50 | +2.62 |
| LDAM [6] | ✓ | 36.05† | +0.17 | 41.86† | +2.98 | 64.58‡ | +3.70 |
| OLTR‡ [42] | ✗ | 35.60 | -0.28 | 40.36 | +1.48 | 63.90 | +3.02 |
| LDAM+DRW [6] | ✗ | 38.22† | +2.34 | 45.75† | +6.87 | 68.00‡ | +7.12 |
| BBN‡ [64] | ✗ | - | | - | | 66.29 | +5.41 |
| c-RT [30] | ✗ | 41.80‡ | +5.92 | 47.54† | +8.66 | 67.60† | +6.72 |
| Ours | ✓ | **42.90** | **+7.02** | **48.41** | **+9.53** | **69.15** | **+8.27** |

**Results.** Tab.3 illustrates the results on large-scale datasets. Ours is consistently effective and outperforms existing mainstream methods, achieving distinguish improvement compared with previous SOTA c-RT [30] in the compared backbones. Especially, our method outperforms the baseline on ImageNet-LT and iNaturalist 2018 by **9.53%** and **8.27%** with ResNet-50, respectively. As can be noticed in Tab.3, the proposed method also surpasses the well-known two-stage methods [30, 6, 64], achieving superior accuracy with less computation load in a concise training manner.

### 4.4 Further Analysis

**Effectiveness of UniMix and Bayias.** We conduct extensive ablation studies in Tab.4 to demonstrate the effectiveness of the proposed UnixMix and Bayias, with detailed analysis in various combinations of feature-wise and loss-wise methods on CIFAR-10-LT and CIFAR-100-LT. Indeed, both UniMix and Bayias turn out to be effective in LT scenarios. Further observation shows that with *calibration* ensured, the proposed method gets significant performance gains and achieve state-of-the-art results. Noteworthy, LDAM [6] makes classifiers miscalibrated, which leads to unsatisfactory improvement when combined with *mixup* manners.

Table 4: Ablation study between feature-wise and loss-wise methods. LDAM is counterproductive to *mixup* and its extensions. Bayias-compensated CE ensures *calibration* and shows excellent performance gains especially combined with UniMix.

| Dataset | | CIFAR-10-LT | | | | CIFAR-100-LT | | | |
|---|---|---|---|---|---|---|---|---|---|
| $\rho$ (easy $\rightarrow$ hard) | | 100 | | 200 | | 100 | | 200 | |
| Mix | Loss | Top1 Acc | $\Delta$ | Top1 Acc | $\Delta$ | Top1 Acc | $\Delta$ | Top1 Acc | $\Delta$ |
| None | CE | 70.36 | - | 66.21 | - | 38.32 | - | 34.56 | - |
| *mixup* | CE | 73.06 | +2.70 | 67.73 | +1.52 | 39.54 | +1.22 | 34.97 | +0.41 |
| Remix | CE | 75.36 | +5.00 | 67.08 | +0.87 | 41.94 | +3.62 | 36.99 | +2.43 |
| UniMix | CE | 76.47 | +6.11 | 68.42 | +2.21 | 41.46 | +3.14 | 37.63 | +3.07 |
| None | LDAM | 74.47 | - | 69.50 | - | 41.76 | - | 37.73 | - |
| *mixup* | LDAM | 73.96 | -0.15 | 67.89 | -1.61 | 40.22 | -1.54 | 37.52 | -0.21 |
| Remix | LDAM | 74.33 | -0.14 | 69.66 | +0.16 | 40.59 | -1.17 | 37.66 | -0.07 |
| UniMix | LDAM | 75.35 | +0.88 | 70.77 | +1.27 | 41.67 | -0.09 | 37.83 | +0.01 |
| None | Bayias | 78.70 | - | 74.21 | - | 43.52 | - | 38.83 | - |
| *mixup* | Bayias | 81.75 | +3.05 | 76.69 | +2.48 | 44.56 | +1.04 | 41.19 | +2.36 |
| Remix | Bayias | 81.55 | +2.85 | 75.81 | +1.60 | 45.01 | +1.49 | 41.44 | +2.61 |
| UniMix | Bayias | **82.75** | **+4.05** | **78.48** | **+4.27** | **45.45** | **+1.93** | **42.07** | **+3.24** |

**Evaluating different UniMix Sampler.** Corollary 1,2,3 demonstrate distinguish influence of UniMix. However, the $\xi$-*sample* can not be completely equivalent with orginal ones. Hence, an appropriate $\tau$ in Eq.9 is also worth further searching. Fig.4 illustrates the accuracy with different $\tau$ on CIFAR-10-LT and CIFAR-100-LT setting $\rho = 10$ and 100. For CIFAR-10-LT (Fig.4(a),4(b)), $\tau = -1$ is possibly ideal, which forces more head-tail instead of head-head pairs get generated to compensate tail classes. In the more challenging CIFAR-100-LT, $\tau = 0$ achieves the best result. We suspect that unlike simple datasets (e.g., CIFAR-10-LT), where overconfidence occurs in head classes, all classes need to get enhanced in complicated LT scenarios. Hence, the augmentation is effective and necessary both on head and tail. $\tau = 0$ allows both head and tail get improved simultaneously.

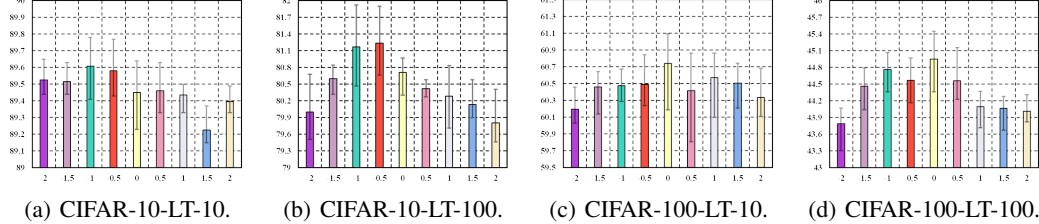

| (a) CIFAR-10-LT-10. | (b) CIFAR-10-LT-100. | (c) CIFAR-100-LT-10. | (d) CIFAR-100-LT-100. |
|---|---|---|---|

Figure 4: Comparison of top-1 validation accuracy(%) of ResNet-32 on CIFAR-LT when varying $\tau$ in Eq.9 for UniMix. The histogram indicates average results in repeated experiments.

**Do minorities really get improved?** To observe the amelioration on tail classes, Fig.5 visualizes log-confusion matrices on CIFAR-100-LT-100. In Fig.5(e), our method exhibits satisfactory generalization on the tail. Vanilla ERM model (Fig.5(a)) is a trivial predictor which simplifies tail instances as head labels to minimize the error rate. Feature improvement [10] and loss modification [6, 58] methods do alleviate LT problem to some extent. The misclassification cases (i.e., non-diagonal elements) in Fig.5(b),5(c),5(d) become smaller and more balanced distributed compared with ERM. However, the error cases are still mainly in the upper or lower triangular, indicating the existence of inherent bias between the head and tail. Our method (Fig.5(e)) significantly alleviates such dilemma. The non-diagonal elements are more uniformly distributed throughout the matrix rather than in

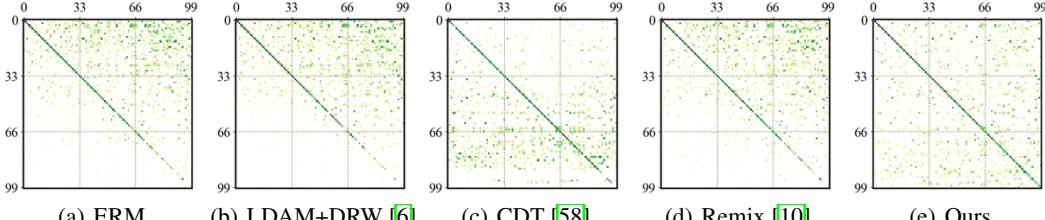

|     (a) ERM     |  (b) LDAM+DRW [6]  |  (c) CDT [58]  |  (d) Remix [10]  |  (e) Ours  |

Figure 5: The log-confusion matrix on CIFAR-100-LT-100 validation dataset. The $x$-axis and $y$-axis indicate the ground truth and predicted labels, respectively. Deeper color indicates larger values.

the corners, showing superiority to erase the bias in LT scenarios. Our method enables effective feature improvement for data-scarce classes and alleviates the *prior* bias, suggesting our success in regularizing tail remarkably.

## 5  Related work and discussion

**Why need *calibration*?**  To quantify the predictive uncertainty, *calibration* [2] is put forward to describe the relevance between predictive score and actual correctness likelihood. A well-calibrated model is more reliable with better interpretability, which probabilities indicate optimal expected costs in Bayesian decision scenarios [43]. Guo *et al.* [16] firstly provide metric to measure the *calibration* of CNN and figure out well-performed models are always in lack of *calibration*, indicating that CNN is sensitive to be overconfidence and lacks robustness. Thulasidasan *et al.* [54] point out that the effectiveness of *mixup* in balanced datasets originates from superior *calibration* modification. Menon *et al.* [44] further show how to ensure optimal classification *calibration* for a pair-wise loss.

**Feature-wise methods.**  Intuitively, under-sampling the head [18, 4, 20] or over-sampling the tail [9, 4, 45, 50, 5] can improve the inconsistent performance of imbalanced datasets but tend to either weaken the head or over-fitting the tail. Hence, many effective works generate additional samples [11, 62, 33] to compensate the tail classes. BBN [64] uses two branches to extract features from head and tail simultaneously, while c-RT [30] trains feature representation learning and classification stage separately. *mixup* [60] and its variants [55, 59, 10] are effective and easy-implement feature-wise methods that convexly combine input and label pairs to generate virtual samples. However, naïve *mixup* manners are deficient in LT scenarios as we discussed in Sec. 2. In contrast, our UniMix tackles such a dilemma by constructing class balance-oriented virtual data as describe in Sec. 3.1 and shows satisfactory *calibration* as Fig. 1 exhibits.

**Loss modification.**  Numerous experimental and theoretical studies [46, 12, 53, 57] have demonstrated the existence of inherent *bias* between LT train set and balanced test set in supervised learning. Previous works [27, 28, 31, 12, 38] make networks prefer learning tail samples by additional class-related weight on CE loss. Some works further correct CE according to the gradient generated by different samples [52, 36] or from the perspective of Gaussian distribution and Bayesian estimation [19, 32]. Meta-learning approaches [15, 1, 51, 47, 29] optimize the weights of each class in CE as learnable parameters and achieve remarkable success. Cao *et al.* [6] theoretically provides the ideal optimal *margin* for CE from the perspective of *VC* generalization bound. Compared with Logit Adjustment [44] motivated by balance error rate, our Bayias-compensated CE eliminates *bias* incured by *prior* and is consistent with balanced datasets, which ensures classification *calibration* as well.

## 6  Conclusion

We systematically analyze the limitations of mainstream feature improvement methods, i.e., *mixup* and its extensions in the label-imbalanced situation, and propose the UniMix to construct a more class-balanced virtual dataset that significantly improves classification *calibration*. We further pinpoint an inherent bias induced by the inconstancy of label distribution *prior* between long-tailed train set and balanced test set. We prove that the standard cross-entropy loss with the proposed Bayias compensated can ensure classification *calibration*. The combination of UniMix and Bayias achieves state-of-the-art performance and contributes to a better-calibrated model (Fig. 1). Further study in Tab. 4 shows that the bad *calibration* methods are counterproductive with each other. However, more in-depth analysis and theoretical guarantees are still required, which we leave for our future work.

## Acknowledgments and Disclosure of Funding

This work was supported by NSFC project Grant No. U1833101, SZSTI Grant No. JCYJ20190809
172201639 and WDZC20200820200655001, the Joint Research Center of Tencent and Tsinghua.

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
