# Towards Calibrated Model for Long-Tailed Visual Recognition from Prior Perspective
# – Supplementary Material –

**Zhengzhuo Xu**[1*]**, Zenghao Chai**[1*]**, Chun Yuan**[1,2†]
[1]Shenzhen International Graduate School, Tsinghua University
[2]Peng Cheng Laboratory
xzz20@mails.tsinghua.edu.cn, zenghaochai@gmail.com,
yuanc@sz.tsinghua.edu.cn

## A  Missing proofs and derivations of UniMix

### A.1  Basic setting

**Setting 1** *Without loss of generality, we suppose that the long-tailed distribution satisfies some kind exponential distribution with parameter $\lambda$ [8]. The imbalance factor is defined as $\rho = n_{\max}/n_{\min}$, where $n_{\max}$ and $n_{\min}$ represent the number of the most and least samples in the dataset with $C$ classes, respectively. Then the probability of class $Y$ belongs to $y_i$ is:*

$$\mathbb{P}(Y = y_i) = \begin{cases} \alpha e^{-\lambda y_i} & y_i \in [1, C] \\ 0 & others \end{cases} \quad \sim \quad long-tailed\ distribution \quad\quad \text{(A.1)}$$

*According to the definition of $\rho$, we can directly deduce the relationship between $\rho$ and $\lambda$:*

$$\rho = \frac{n_{\max}}{n_{\min}} = \frac{\mathbb{P}(Y = y_i)_{\max}}{\mathbb{P}(Y = y_i)_{\min}} \Rightarrow \frac{\mathbb{P}(Y = y_1)}{\mathbb{P}(Y = y_C)} = \frac{\alpha e^{-\lambda \cdot 1}}{\alpha e^{-\lambda \cdot C}} \quad \Rightarrow \quad \lambda = \frac{\ln \rho}{C - 1} \quad\quad \text{(A.2)}$$

*Considering the normalization of probability density, we have:*

$$\int_{y_i \in \mathcal{Y}} \mathbb{P}(Y = y_i) dy_i = \int_1^C \alpha e^{-\lambda y_i} dy_i \equiv 1$$
$$\Rightarrow -\frac{\alpha}{\lambda} e^{-\lambda y_i} \Big|_1^C \equiv 1 \Rightarrow \frac{\alpha}{\lambda} \left( e^{-\lambda} - e^{-\lambda C} \right) \equiv 1 \quad\quad \text{(A.3)}$$
$$\Rightarrow \alpha = \frac{\lambda}{e^{-\lambda} - e^{-\lambda C}}$$

*Hence, we can express the distribution of LT dataset $\mathcal{D}_{train}$, i.e., $\mathbb{P}(Y = y_i)$ represents the probability of class $y_i$ in $\mathcal{D}_{train}$, which can be calculated as follows:*

$$\mathbb{P}(Y = y_i) = \frac{\iint_{x_i \in \mathcal{X}, y_j \in \mathcal{Y}} \mathbb{1}(X = x_i, Y = y_i) dx_i dy_j}{\iint_{x_i \in \mathcal{X}, y_j \in \mathcal{Y}} \mathbb{1}(X = x_i, Y = y_j) dx_i dy_j} = \frac{\lambda}{e^{-\lambda} - e^{-\lambda C}} e^{-\lambda y_i}, y_i \in [1, C] \quad \text{(A.4)}$$

*Notice that Eq.A.4 is determined by total class number $C$ and the imbalance factor $\rho$.*

### A.2  Proof of Corollary 1

**Corollary 1** *When $\xi \sim Beta(\alpha, \alpha), \alpha \in [0, 1]$, the newly mixed dataset $\mathcal{D}_\nu$ composed of $\xi$-Aug samples $(\widetilde{x}_{i,j}, \widetilde{y}_{i,j})$ follows the same long-tailed distribution as the origin dataset $\mathcal{D}_{train}$, where*

---

[*]Equal contribution.
[†]Corresponding author.

35th Conference on Neural Information Processing Systems (NeurIPS 2021).

$(x_i, y_i)$ and $(x_j, y_j)$ are **randomly** sampled from $\mathcal{D}_{train}$.

$$\mathbb{P}_{mixup}(Y^* = y_i) = \mathbb{P}^2(Y = y_i) + \mathbb{P}(Y = y_i) \iint_{y_i \neq y_j} Beta(\alpha, \alpha)\mathbb{P}(Y = y_j)d\xi dy_j$$

$$= \frac{\lambda}{e^{-\lambda} - e^{-\lambda C}}e^{-\lambda y_i}, y_i \in [1, C] \tag{A.5}$$

**Proof A.1** *Follow the Basic Setting 1, when mixing factor $\xi \sim Beta(\alpha, \alpha)$, consider a $\xi$-Aug sample generated by $\widetilde{x}_{i,j} = \xi \cdot x_i + (1 - \xi) \cdot x_j$ and $\widetilde{y}_{i,j} = \xi \cdot y_i + (1 - \xi) \cdot y_j$. The $\xi$-Aug sample $\widetilde{x}_{i,j}$ contributes to class $y_i$ when both pair samples $(x_i, y_i)$ and $(x_j, y_j)$ are in class $y_i$ or one of them are in other labels while the mixing factor $\xi$ is in favor of class $y_i$.*

$$\mathbb{P}(\tilde{y}_{i,j} = y_i) = \mathbb{P}(\tilde{y}_{i,j} = y_i) \cdot \mathbb{P}(\tilde{y}_{i,j} = y_i)\mathbb{P}_\xi(0 \leq \xi \leq 1)$$
$$+ \mathbb{P}(\tilde{y}_{i,j} = y_i) \cdot \mathbb{P}(\tilde{y}_{i,j} \neq y_i)\mathbb{P}_\xi(\xi \geq 0.5) \tag{A.6}$$
$$+ \mathbb{P}(\tilde{y}_{i,j} \neq y_i) \cdot \mathbb{P}(\tilde{y}_{i,j} = y_i)\mathbb{P}_\xi(\xi < 0.5)$$

*Hence, the distribution of new dataset $D_\nu$ is:*

$$\mathbb{P}_{mixup}(Y = y_i) = \mathbb{P}(Y = y_i) \cdot \mathbb{P}(Y = y_i) \cdot \int_{0 \leq \xi \leq 1} Beta(\alpha, \alpha)d\xi$$

$$+ \mathbb{P}(Y = y_i) \cdot \int_{y_j \neq y_i} \int_{\xi \geq 0.5} Beta(\alpha, \alpha) \cdot \mathbb{P}(Y = y_j)d\xi dy_j$$

$$+ \int_{y_j \neq y_i} \int_{\xi < 0.5} Beta(\alpha, \alpha) \cdot \mathbb{P}(Y = y_j)d\xi dy_j \cdot \mathbb{P}(Y = y_i)$$
$$= \mathbb{P}^2(Y = y_i) + 0.5 \cdot \mathbb{P}(Y = y_i) \cdot (1 - \mathbb{P}(Y = y_i)) \tag{A.7}$$
$$+ 0.5 \cdot (1 - \mathbb{P}(Y = y_i)) \cdot \mathbb{P}(Y = y_i)$$
$$= \mathbb{P}^2(Y = y_i) + 0.5 \cdot \mathbb{P}(Y = y_i) - 0.5 \cdot \mathbb{P}^2(Y = y_i)$$
$$+ 0.5 \cdot \mathbb{P}(Y = y_i) - 0.5 \cdot \mathbb{P}^2(Y = y_i)$$
$$= \mathbb{P}(Y = y_i) = \frac{\lambda}{e^{-\lambda} - e^{-\lambda C}}e^{-\lambda y_i}$$

*According to Eq.A.4 and Eq.A.7, the $\xi$-Aug samples in mixed dataset $\mathcal{D}_\nu$ generated by mixup follow the same distribution of the original long-tailed one. Therefore, the head gets more regulation than the tail. One the one hand, the classification performance will be promoted. On the other hand, however, the performance gap between the head and tail still exists.*

## A.3 Proof of Corollary 2

**Corollary 2** *When $\xi_{i,j}^* \sim \mathcal{U}(\pi_{y_i}, \pi_{y_j}, \alpha, \alpha), \alpha \in [0, 1]$, the newly mixed dataset $\mathcal{D}_\nu$ composed of $\xi$-Aug samples $(\widetilde{x}_{i,j}, \widetilde{y}_{i,j})$ follows a middle-majority distribution, where $(x_i, y_i)$ and $(x_j, y_j)$ are both **randomly** sampled from $\mathcal{D}_{train}$.*

$$\mathbb{P}_{mixup}^*(Y^* = y_i) = \mathbb{P}(Y = y_i)\int_{y_j < y_i} \mathbb{1}\left(\int \xi_{i,j}^* \mathcal{U}(\pi_i, \pi_j, \alpha, \alpha)d\xi_{i,j}^* \geq 0.5\right)\mathbb{P}(Y = y_j)dy_j$$

$$= \frac{\lambda}{(e^{-\lambda} - e^{-C\lambda})^2}\left(e^{-\lambda(y_i+1)} - e^{-2\lambda y_i}\right), y_i \in [1, C] \tag{A.8}$$

**Proof A.2** *Follow the settings of Proof A.2 and Eq.6, we can get the following relationship of the label $y_i$ and $y_j$ with UniMix Factor $\xi_{i,j}^*$. It easy to know $\pi_{y_i} \leq \pi_{y_j}$ if the index $i \geq j$ for the reason that class $y_j$ occupies more instances than class $y_i$ in long-tailed distribution described as Eq.A.4. Under this circumstances, if consider UniMix Factor $\xi_{i,j}^*$ as a constant intuitively, we can deduce that $\xi_{i,j} = \pi_{y_j}/(\pi_{y_i} + \pi_{y_j}) \geq 0.5$, i.e.,*

$$i \geq j \quad \Rightarrow \quad \pi_{y_i} \leq \pi_{y_j} \quad \Rightarrow \quad \xi_{i,j} = \frac{\pi_{y_i}}{\pi_{y_i} + \pi_{y_j}} \geq 0.5 \tag{A.9}$$

*To improve the robustness and generalization, we consider $\xi_{i,j}^* \sim \mathscr{U}(\pi_{y_i}, \pi_{y_j}, \alpha, \alpha)$, which extends $\xi_{i,j}^*$ to $\pi_{y_j}/(\pi_{y_i} + \pi_{y_j})$ and its vicinity. Hence we can generalize Eq.A.9 to:*

$$y_i \geq y_j \quad \Rightarrow \quad \pi_{y_i} \leq \pi_{y_j} \quad \Rightarrow \quad \xi_{i,j} \geq 0.5 \Rightarrow \int \xi_{i,j}^* \mathscr{U}(\pi_i, \pi_j, \alpha, \alpha) d\xi_{i,j}^* \geq 0.5 \quad \text{(A.10)}$$

*Given any $i, j \in [1, C]$, the $\widetilde{x}_{i,j}$ will tend to be a $\xi$-Aug sample of the class $y_i$ if $i \geq j$ for the tail-favored mixing factor. In other words, $\widetilde{x}_{i,j}$ will be a $\xi$-Aug sample for class $y_i$ with the mean of $\xi_{i,j}^*$ to be $\pi_{y_j}/(\pi_{y_i} + \pi_{y_j})$. Hence, the probability that $\widetilde{x}_{i,j}$ belongs to class $y_i$ is:*

$$
\begin{aligned}
\mathbb{P}_{mixup}^*(Y = y_i) &= \mathbb{P}(Y = y_i) \cdot \mathbb{1}\left(\int \xi_{i,j}^* \mathscr{U}(\pi_i, \pi_j, \alpha, \alpha) d\xi_{i,j}^* \geq 0.5\right) \cdot \mathbb{P}(Y < y_i) \\
&= \mathbb{P}(Y = y_i) \int_{y_j < y_i} \mathbb{1}\left(\int \xi_{i,j}^* \mathscr{U}(\pi_i, \pi_j, \alpha, \alpha) d\xi_{i,j}^* \geq 0.5\right) \mathbb{P}(Y = y_j) dy_j \\
&= \mathbb{P}(Y = y_i) \cdot \int_1^{y_i} \mathbb{P}(Y = y_j) dy_j \\
&= \frac{\lambda}{e^{-\lambda} - e^{-\lambda C}} e^{-\lambda y_i} \cdot \int_1^{y_i} \frac{\lambda}{e^{-\lambda} - e^{-\lambda C}} e^{-\lambda y_j} dy_j \\
&= \frac{\lambda}{e^{-\lambda} - e^{-\lambda C}} e^{-\lambda y_i} \cdot \left. -\frac{\frac{\lambda}{e^{-\lambda} - e^{-\lambda C}}}{\lambda} e^{-\lambda y_j} \right|_1^{y_i} \\
&= \frac{\lambda}{(e^{-\lambda} - e^{-C\lambda})^2} \left(e^{-\lambda(y_i+1)} - e^{-2\lambda y_i}\right)
\end{aligned}
$$
$$\text{(A.11)}$$

*According to Eq.A.11, it's easy to find a derivative zero point in range [1,C]. Hence, the newly distributed dataset $\mathcal{D}_\nu$ generated by mixup with UniMix Factor $\xi_{i,j}^*$ follows a middle-majority distribution that most data concentrates on middle classes.*

## A.4 Proof of Corollary 3

**Corollary 3** *When $\xi_{i,j}^* \sim \mathscr{U}(\pi_{y_i}, \pi_{y_j}, \alpha, \alpha), \alpha \in [0, 1]$, the newly mixed dataset $\mathcal{D}_\nu$ composed of $\xi$-Aug samples $(\widetilde{x}_{i,j}, \widetilde{y}_{i,j})$ follows a tail-majority distribution, where $(x_i, y_i)$ is **randomly** and $(x_j, y_j)$ is **inversely** sampled from $\mathcal{D}_{train}$, respectively.*

$$
\begin{aligned}
\mathbb{P}_{UniMix}(Y^* = y_i) &= \mathbb{P}(Y = y_i) \int_{y_j < y_i} \mathbb{1}\left(\int \xi_{i,j}^* \mathscr{U}(\pi_i, \pi_j, \alpha, \alpha) d\xi_{i,j}^* \geq 0.5\right) \mathbb{P}_{inv}(Y = y_j) dy_j \\
&= \frac{\lambda}{(e^{-\lambda} - e^{-C\lambda})(e^{-C\tau\lambda} - e^{-\tau\lambda})} \left(e^{-\lambda y_i(\tau+1)} - e^{-\lambda(\tau+y_i)}\right), y_i \in [1, C]
\end{aligned}
$$
$$\text{(A.12)}$$

**Proof A.3** *Follow the Basic Setting of Proof.1. suppose $(x_i, y_i)$ is randomly sampled from $\mathcal{D}_{train}$, while $(x_j, y_j)$ is inversely sampled from $\mathcal{D}_{train}$ related to $\tau$. i.e., the probability of $\mathbb{P}(Y = y_i)$ is:*

$$\mathbb{P}(Y = y_i) = \frac{\lambda}{e^{-\lambda} - e^{-\lambda C}} e^{-\lambda y_i} \quad \text{(A.13)}$$

*The probability of UniMix Sampler that the sampled class $Y$ belongs to $y_i$ is $\mathbb{P}_{inv}(Y = y_j)$:*

$$
\begin{aligned}
\mathbb{P}_{inv}(Y = y_j) &= \frac{\mathbb{P}^\tau(Y = y_j)}{\int_{y_k \in \mathcal{Y}} \mathbb{P}^\tau(Y = y_k) dy_k} \\
&= \frac{\left(\frac{\lambda}{e^{-\lambda} - e^{-\lambda C}} e^{-\lambda y_j}\right)^\tau}{\int_{y_k \in \mathcal{Y}} \left(\frac{\lambda}{e^{-\lambda} - e^{-\lambda C}} e^{-\lambda y_k}\right)^\tau dy_k} \\
&= \frac{e^{-\lambda \tau y_j}}{\int_1^C e^{-\lambda \tau y_k} dy_k} = \frac{\lambda \tau e^{-\lambda \tau y_j}}{e^{-\lambda \tau} - e^{-\lambda \tau C}}
\end{aligned}
$$
$$\text{(A.14)}$$

*Follow the same proof as Proof.A.3, the probability that $\widetilde{x}_{i,j}$ belongs to class $y_i$ is:*

$$
\begin{aligned}
\mathbb{P}_{UniMix}(Y = y_i) &= \mathbb{P}(Y = y_i) \cdot \mathbb{1}\left(\int \xi_{i,j}^* \mathscr{U}(\pi_i, \pi_j, \alpha, \alpha) d\xi_{i,j}^* \geq 0.5\right) \cdot \mathbb{P}_{inv}(Y < y_i) \\
&= \mathbb{P}(Y = y_i) \int_{y_j < y_i} \mathbb{1}\left(\int \xi_{i,j}^* \mathscr{U}(\pi_i, \pi_j, \alpha, \alpha) d\xi_{i,j}^* \geq 0.5\right) \mathbb{P}_{inv}(Y = y_j) dy_j \\
&= \mathbb{P}(Y = y_i) \cdot \int_1^{y_i} \mathbb{P}_{inv}(Y = y_j) dy_j \\
&= \frac{\lambda}{e^{-\lambda} - e^{-\lambda C}} e^{-\lambda y_i} \cdot \int_1^{y_i} \frac{\lambda \tau e^{-\lambda \tau y_j}}{e^{-\lambda \tau} - e^{-\lambda \tau C}} dy_j \\
&= \frac{\lambda}{e^{-\lambda} - e^{-\lambda C}} e^{-\lambda y_i} \cdot \frac{-e^{-\lambda \tau y_j}}{e^{-\lambda \tau} - e^{-\lambda \tau C}}\bigg|_1^{y_i} \\
&= \frac{\lambda}{\left(e^{-\lambda} - e^{-C\lambda}\right)\left(e^{-C\tau\lambda} - e^{-\tau\lambda}\right)}\left(e^{-\lambda y_i(\tau+1)} - e^{-\lambda(\tau+y_i)}\right)
\end{aligned}
$$

$$(A.15)$$

*According to Eq.A.15, $\mathbb{P}_{UniMix}(Y = y_i)$ is a gently increasing function in range [1,C]. Hence, the newly mixed dataset $\mathcal{D}_\nu$ generated by UniMix follows a tail-majority distribution that adequate data concentrates on tail classes.*

## A.5 More visualization of Corollary 1,2,3

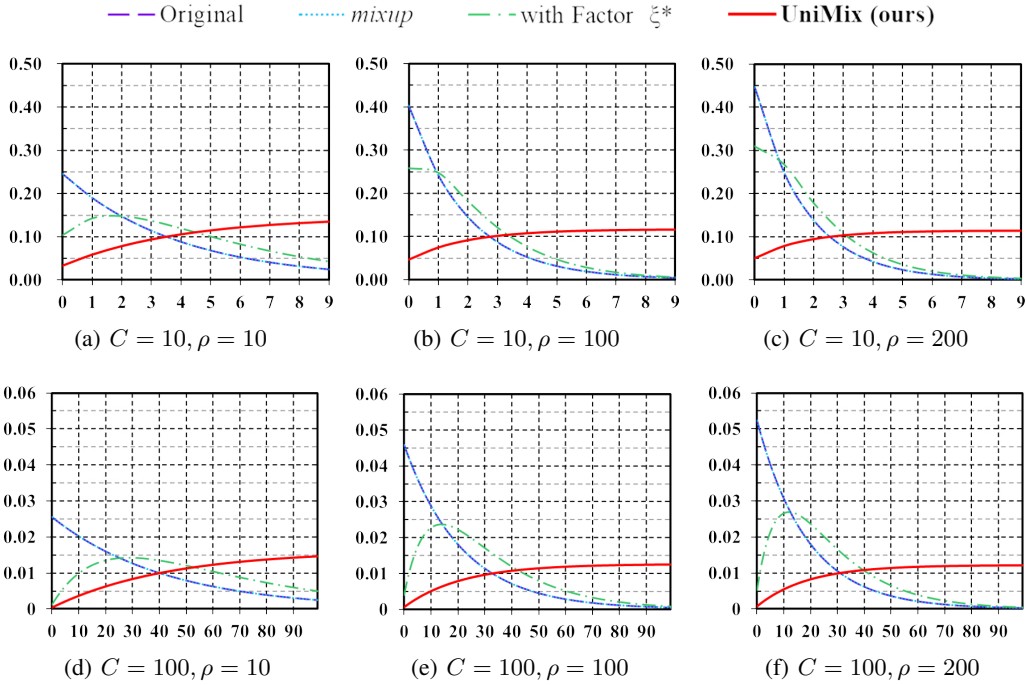

Figure A1: Additional visualized comparisons of $\xi$-*Aug* samples distribution in Corollary 1,2,3. $x$-axis: class indices. $y$-axis: probability of each class. *mixup* (blue) exhibits the same LT distribution as origin (purple). UniMix Factor (green) alleviates such situation and the full pipeline ($\tau{=}{-}1$) constructs a more uniform distribution of $\xi$-*Aug* (red), which contributes to a well-calibrated model.

We present additional visualized distribution of $\xi$-*Aug* samples with different sample strategies for comprehensive comparisons, including $\rho \in \{10, 100, 200\}$ and $C \in \{10, 100\}$. As illustrated in Fig.A1, when the class number $C$ and imbalance factor $\rho$ get larger, the limitations of *mixup* in LT scenarios gradually appear. The VRM dataset $\mathcal{D}_\nu$ generated by *mixup* with naïve mixing factor and random sampler will make $\mathcal{D}_\nu$ following the same LT distribution. The newly mixed pseudo data by

*mixup* follows the head-majority distribution. It has limited contribution for the tail class' feature learning and regulation, which is the reason for its poor *calibration*.

In contrast, the proposed UniMix Factor $\xi_{i,j}^*$ significantly promotes such situation and improves tail class's feature learning by the preference on the relatively fewer classes of the two samples in a pair. Because of the random sampler, the samples are still mainly from the head, and the newly mixed dataset will follows a middle-majority distribution. Most data concentrates on the middle of classes as green line shows. As $C$ and $\rho$ get larger, the middle distribution will get close to the head. Thanks to the UniMix Sampler that inversely draws data from $\mathcal{D}_{train}$, the $D_\nu$ is mainly composed of the head-tail pairs that contribute to the feature learning for the tail when integrated with UniMix Factor. As a result, $D_\nu$ follows the tail-majority distribution that improves the generalization on the tail. Specifically, when $C$ and $\rho$ get larger, the distribution of $D_\nu$ generated by UniMix still maintains satisfactory tail-majority distribution.

# B  Missing Proofs and derivations of Bayias

## B.1  Proof of Bayias

**Theorem B.1** *For classification, let $\psi(x; \theta, W, b)$ be a hypothesis class of neural networks of input $X = x$, the classification with Softmax should contain the influence of prior, i.e., the predicted label during training should be:*

$$\hat{y} = \arg\max_{y_i \in \mathcal{Y}} \frac{e^{\psi(x;\theta,W,b)_{y_i} + \log(\pi_{y_i}) + \log(C)}}{\sum_{y_j \in \mathcal{Y}} e^{\psi(x;\theta,W,b)_{y_j} + \log(\pi_{y_j}) + \log(C)}} \tag{B.1}$$

**Proof B.1** *Generally, a classifier can be modeled as:*

$$\hat{y} = \arg\max_{y_i \in \mathcal{Y}} \frac{e^{\sum_{d_i \in D}[(W^T)_{y_i}^{(d_i)} \mathcal{F}(x;\theta)^{(d_i)}] + b_{y_i}}}{\sum_{y_j \in \mathcal{Y}} e^{\sum_{d_i \in D}[(W^T)_{y_j}^{(d_i)} \mathcal{F}(x;\theta)^{(d_i)}] + b_{y_j}}} \triangleq \arg\max_{y_i \in \mathcal{Y}} \frac{e^{\psi(x;\theta,W,b)_{y_i}}}{\sum_{y_j \in \mathcal{Y}} e^{\psi(x;\theta,W,b)_{y_j}}} \tag{B.2}$$

*where $\hat{y}$ indicates the predicted label and $\mathcal{F}(x;\theta) \in \mathbb{R}^{D \times 1}$ is the D-dimension feature extracted by the backbone with parameter $\theta$. $W \in \mathbb{R}^{D \times C}$ represents the parameter matrix of the classifier. The model attempts to get the maximum $\hat{y}$ given $x$, i.e., to maximize the posterior, satisfying the following relationship between the prior and likelihood according to the Bayesian theorem:*

$$\begin{aligned}
\hat{y} &= \arg\max_{y_i \in \mathcal{Y}} \mathbb{P}(Y = y_i | X = x) \\
&= \arg\max_{y_i \in \mathcal{Y}} \frac{\mathbb{P}(X = x | Y = y_i) \cdot \mathbb{P}(Y = y_i)}{\sum_k \mathbb{P}(Y = y_k) \prod_j \mathbb{P}(X^{(j)} = x^{(j)} | Y = y_k)} \\
&\propto \arg\max_{y_i \in \mathcal{Y}} \mathbb{P}(X = x | Y = y_i) \cdot \mathbb{P}(Y = y_i)
\end{aligned} \tag{B.3}$$

*where $\sum_k \mathbb{P}(Y = y_k) \prod_j \mathbb{P}(X^{(j)} = x^{(j)} | Y = y_k)$ is the normalized evidence factor. $\mathbb{P}(Y = y_i)$ is the prior probability estimated by the instance proportion of each category in the dataset. To maximize posterior, we need to get the $\mathbb{P}(X|Y)$ in an ERM supervised training manner. However, in LT scenarios, the likelihood is consistent in the train and test set, but the prior is different. Hence, we derive the posterior on train and test set separately:*

$$\begin{cases}
\hat{y} \propto \arg\max_{y_i \in \mathcal{Y}} \mathbb{P}(X = x | Y = y_i) \cdot \mathbb{P}_{train}(Y = y_i) \\
\hat{y}' \propto \arg\max_{y_i \in \mathcal{Y}} \mathbb{P}(X = x' | Y' = y_i) \cdot \mathbb{P}'_{test}(Y' = y_i)
\end{cases} \tag{B.4}$$

*where $\hat{y}, \hat{y}'$ represent the prediction results for the train and test set, respectively. The model $\psi(x; \theta, W, b)$ is just the likelihood estimation $\mathbb{P}(X = x | Y = y_i)$ of the train set. To obtain the posterior probability $y' = \arg\max_{y_i \in \mathcal{Y}} \mathbb{P}'(Y = y_i | X = x')$ for inference, one should consider unifying the optimization direction on the train set and test set:*

$$\mathbb{P}_{test}(Y = y_i | X = x') \propto \frac{\mathbb{P}_{train}(Y = y_i | X = x)}{\mathbb{P}_{train}(Y = y_i)} \cdot \mathbb{P}_{test}(Y' = y_i) = \frac{\mathbb{P}_{train}(Y = y_i | X = x)}{C \cdot \pi_{y_i}} \tag{B.5}$$

*For the difference of label prior, the learned parameters of the model will also yield class-level bias. Hence, the actual optimization direction is not described as Eq.B.2 because the bias incurred by prior should be compensated at first.*

$$
\begin{aligned}
\hat{\theta}, \hat{W}, \hat{b} \triangleq \Theta &= \underset{\Theta}{\arg\min} \sum_{\substack{x_i \in \mathcal{X}' \\ y_i \in \mathcal{Y}}} \mathbb{1} \left( y_i \neq \underset{y_j \in \mathcal{Y}}{\arg\max} (\mathbb{P}_{test}(Y = y_j | X = x_i)) \right) \\
&\Leftrightarrow \underset{\Theta}{\arg\min} \sum_{\substack{x_i \in \mathcal{X} \\ y_i \in \mathcal{Y}}} \mathbb{1} \left( y_i \neq \underset{y_j \in \mathcal{Y}}{\arg\max} \frac{\mathbb{P}_{train}(Y = y_j | X = x_i)}{C \cdot \pi_{y_j}} \right) \\
&= \underset{\Theta}{\arg\min} \sum_{\substack{x_i \in \mathcal{X} \\ y_i \in \mathcal{Y}}} \mathbb{1} \left( y_i \neq \underset{y_j \in \mathcal{Y}}{\arg\max} \frac{e^{\psi(x_i;\theta,W,b)_{y_j} - \log C - \log(\boldsymbol{\pi_{y_j}})}}{\sum_{y_k \in \mathcal{Y}} e^{\psi(x_i;\theta,W,b)_{y_k}}} \right) \\
&\Leftrightarrow \underset{\Theta}{\arg\min} \sum_{\substack{x_i \in \mathcal{X} \\ y_i \in \mathcal{Y}}} \mathbb{1} \left( y_i \neq \underset{y_j \in \mathcal{Y}}{\arg\max} \frac{e^{\psi(x_i;\theta,W,b)_{y_j} - \log(\boldsymbol{\pi_{y_j}}) - \log C}}{\sum_{y_k \in \mathcal{Y}} e^{\psi(x_i;\theta,W,b)_{y_k} - \log(\boldsymbol{\pi_{y_k}}) - \log C}} \right)
\end{aligned}
\tag{B.6}
$$

*To correct the bias for inferring, the offset term that the model in LT datasets needs to compensate is:*

$$
\mathscr{B}_y = log(\pi_y) + log(C)
\tag{B.7}
$$

## B.2 Proof of classification *calibration*

**Theorem B.2** $\mathscr{B}_y$-*compensated cross-entropy loss in Eq.B.8 ensures classification calibration.*

$$
\mathcal{L}_{\mathscr{B}}(y_i, \psi(x; \Theta)) = -\log \frac{e^{\psi(x;\Theta)_{y_i} + \log(\pi_{y_i}) + \log(C)}}{\sum_{y_k \in \mathcal{Y}} e^{\psi(x;\Theta)_{y_j} + \log(\pi_{y_k}) + \log(C)}}
\tag{B.8}
$$

Classification *calibration* [10, 26] represents the predicted winning *Softmax* scores indicate the actual likelihood of a correct prediction. The miscalibrated models tend to be overconfident which results in that the minimiser of the expected loss (equally, the empirical risk in the infinite sample limit) can not lead to a minimal classification error. Previous work [21] has introduced a theory to measure the *calibration* of a pair-wise loss:

**Lemma B.1** *Pairwise loss* $\mathcal{L}(y_i, \psi(x; \Theta)) = \alpha_{y_i} \cdot \log \left[ 1 + \sum_{y_k \neq y_i} e^{\Delta_{y_i y_k}} \cdot e^{(\psi(x;\Theta)_{y_i} - \psi(x,\Theta)_{y_k})} \right]$ *ensures classification calibration if for any* $\delta \in \mathbb{R}_+^C$:

$$
\alpha_{y_i} = \delta_{y_i} / \pi_{y_i} \quad \Delta_{y_i} y_k = \log(\delta_{y_k} / \delta_{y_i})
\tag{B.9}
$$

**Proof B.2** *To begin, we rewritten the Bayias-compensated cross-entropy loss into the following equation:*

$$
\begin{aligned}
\mathcal{L}_{\mathscr{B}}(y_i, \psi(x; \Theta)) &= -\log \frac{e^{\psi(x;\Theta)_{y_i} + \log(\pi_{y_i}) + \log(C)}}{\sum_{y_k \in \mathcal{Y}} e^{\psi(x;\Theta)_{y_j} + \log(\pi_{y_k}) + \log(C)}} \\
&\Leftrightarrow \log \frac{\sum_{y_k \in \mathcal{Y}} e^{\psi(x;\Theta)_{y_k} + \log(\pi_{y_k}) + \log(C)}}{e^{\psi(x;\Theta)_{y_i} + \log(\pi_{y_i}) + \log(C)}} \\
&= \log \left[ 1 + \frac{\sum_{y_k \neq y_i} e^{\psi(x;\Theta)_{y_k} + \log(\pi_{y_k}) + \log(C)}}{e^{\psi(x;\Theta)_{y_i} + \log(\pi_{y_i}) + \log(C)}} \right] \\
&= \log \left[ 1 + \sum_{y_k \neq y_i} \frac{e^{\psi(x;\Theta)_{y_k} + \log(\pi_{y_k}) + \log(C)}}{e^{\psi(x;\Theta)_{y_i} + \log(\pi_{y_i}) + \log(C)}} \right] \\
&= \log \left[ 1 + \sum_{y_k \neq y_i} \frac{e^{\log(\pi_{y_k}) + \log(C)} \cdot e^{\psi(x;\Theta)_{y_k}}}{e^{\log(\pi_{y_i}) + \log(C)} \cdot e^{\psi(x;\Theta)_{y_i}}} \right] \\
&= \log \left[ 1 + \sum_{y_k \neq y_i} e^{\mathscr{B}_{y_k} - \mathscr{B}_{y_i}} \cdot e^{\psi(x;\Theta)_{y_k} - \psi(x;\Theta)_{y_i}} \right]
\end{aligned}
\tag{B.10}
$$

*Compare Eq.B.10 with Lemma B.1, observed that when $\delta_y = \pi_y$.*

$$\begin{cases} \alpha_{y_i} = \delta_{y_i}/\pi_{y_i} = \pi_{y_i}/\pi_{y_i} = 1 \\ \Delta_{y_i y_k} = \mathscr{B}_{y_k} - \mathscr{B}_{y_i} \\ \qquad = [log(\pi_{y_k}) + log(C)] - [log(\pi_{y_i}) + log(C)] \\ \qquad = log(\pi_{y_k}) - log(\pi_{y_i}) = log(\pi_{y_k}/\pi_{y_i}) \end{cases} \tag{B.11}$$

*According to Lemma B.1, we immediately deduce that Bayias-compensated cross-entropy loss ensures classification calibration.*

## B.3 Comparisons with other losses

Previous work [16, 8, 25, 30, 4, 21] adjusts the logits *weight* or *margin* on standard *Softmax* cross-entropy (CE) loss to tackle the long-tailed datasets. We summarize the loss modification methods reported in Tab.1 and discuss the difference between theirs and ours here.

**Weight-wise losses.** Focal loss [18] is proposed to balance the positive/negative samples during object detection and extends for classification by assigning low weight loss for easy samples. It re-wights the loss with a factor $(1 - p_{y_i})^\gamma$ on standard cross-entropy loss, where $p_{y_i}$ is the prediction probability of class $y_i$:

$$\mathcal{L}_{Focal} = -(1 - p_{y_i})^\gamma \log \frac{e^{\psi(x;\Theta)_{y_i}}}{\sum_{y_k \in \mathcal{Y}} e^{\psi(x;\Theta)_{y_k}}} \tag{B.12}$$

CB loss is proposed by Cui *et al.* [8] with the *effective number*. It adopts a coefficient $(1 - \beta)/(1 - \beta^{n_{y_i}})$ for standard cross-entropy loss:

$$\mathcal{L}_{CB} = -\frac{1 - \beta}{1 - \beta^{n_{y_i}}} \log \frac{e^{\psi(x;\Theta)_{y_i}}}{\sum_{y_k \in \mathcal{Y}} e^{\psi(x;\Theta)_{y_k}}} \tag{B.13}$$

CDT loss [30] re-weights each *Softmax* logit with a temperature scale factor $a_{y_i} = \left(\frac{n_{max}}{n_{y_i}}\right)^\gamma$. It artificially reduces the decision values for head classes:

$$\mathcal{L}_{CDT} = -\log \frac{e^{\psi(x;\Theta)_{y_i}/(\frac{n_{max}}{n_{y_i}})^\gamma}}{\sum_{y_k \in \mathcal{Y}} e^{\psi(x;\Theta)_{y_k}/(\frac{n_{max}}{n_{y_k}})^\gamma}} \tag{B.14}$$

All above re-weight methods are proven effective empirically, more or less. However, such approaches will confront the coverage dilemma when the train data gets highly imbalanced. The weights related to instances number may be large in this situation and result in unstable gradients that deteriorate head classes' performance gain. They are also sensitive to hyper-parameters, which makes it hard to adopt in varied datasets.

**Margin-wise losses.** Previous work in Deep Metrics Learning [20, 9, 28, 19] attempts to obtain better inter-class and intra-class distance from the margin perspective. Cao *et al.* [4] analyze the optimal margin between two different classes via generalization error bounds in the long tail visual recognition. They propose the LDAM loss to encourage the tail classes to enjoy larger margins. In detail, a label-aware margin $C/n_{y_i}^{1/4}$ is added to the groud truth logit where $C$ is an independent constant:

$$\mathcal{L}_{LDAM} = -\log \frac{e^{\psi(x;\Theta)_{y_i} - C/n_{y_i}^{1/4}}}{e^{\psi(x;\Theta)_{y_i} - C/n_{y_i}^{1/4}} + \sum_{y_k \neq y_i} e^{\psi(x;\Theta)_{y_k}}} \tag{B.15}$$

Logit Adjustment [21] loss is proposed to overcome the long-tailed dataset motivated by the balanced error rate. They suppose that the model shows similar performance on each class in the validation dataset. The authors propose a margin $\tau \log(\pi_{y_i})$ on standard *Softmax* cross-entropy loss. Different from the LDAM loss, such margin is added for all logits with label priors $\pi_y$:

$$\mathcal{L}_{LA} = -\log \frac{e^{\psi(x;\Theta)_{y_i} + \tau \log(\pi_{y_i})}}{\sum_{y_k \in \mathcal{Y}} e^{\psi(x;\Theta)_{y_k} + \tau \log(\pi_{y_k})}} \tag{B.16}$$

Our Bayias-compensated CE loss is motivated by erasing the difference of label *prior* in the train set and test set based on the Bayesian theory. The network parameters are only the *likelihood* estimation, while we need a reliable *posterior* for unbiased inference. Hence, we compensate the bias incurred by various label *prior* via adding a margin related to the train set label *prior* and minus a margin related to the test set label *prior*:

$$\mathcal{L}_{ours} = -\log \frac{e^{\psi(x;\Theta)_{y_i} + \log(\pi_{y_i}) - \log(1/C)}}{\sum_{y_k \in \mathcal{Y}} e^{\psi(x;\Theta)_{y_k} + \log(\pi_{y_k}) - \log(1/C)}} \tag{B.17}$$

One may notice that our loss is similar to LA loss with an extra constant margin $-\log(1/C)$ when the $\tau = 1$ in LA loss. However, the difference lies in two aspects: 1) the margin $-\log(1/C)$ is a positive value and makes all logits get larger, which makes *Softmax* operation more distinguishable. 2) Notice our two margins are related to the label *prior* of the train set and test set. Hence, when the train set is balanced, i.e., $\pi_{y_i} = 1/C$, the margin will be $\log(1/C) - \log(1/C) \equiv 0$, and our loss coverage to the standard cross-entropy loss. Furthermore, our loss can handle the situation that both train set and test set are imbalanced distribution via directly setting the margin as $\log(\pi_{y_i}) - \log(\pi'_{y_i})$, where $\pi_{y_i}$ is the label *prior* in the train set and $\pi'_{y_i}$ is the label *prior* in test set correspondingly.

## C    Implement detail

We conduct experiments on CIFAR-10-LT [17], CIFAR-100-LT [17], ImageNet-LT [24], and iNaturalist 2018 [14]. We adopt long-tailed version CIFAR datasets which are build by suitably discarding training instances following the Exp files given in [8, 4]. The instance number of each class exponentially decays in train dataset and keeps balanced during validation process. Here, an imbalance factor $\rho$ is define as $n_{max}/n_{min}$ to measure how imbalanced the dataset is. ImageNet-LT is the LT version of ImageNet. It samples instances for each class following *Pareto* distribution which is also long-tailed. iNaturalist 2018 is a real-world dataset with $8,142$ classes and $\rho = 500$, which suffers extremely class-imbalanced and fine-grained problems.

### C.1    Implement details on CIFAR-LT

ResNet-32 is the backbone on CIFAR-LT. For all CIFAR-10-LT and CIFAR-100-LT, the universal data augmentation strategies [11] are used for training. Specifically, we pad 4 pixels on each side and crop a $32 \times 32$ region. The cropped regions are flipped horizontally with $0.5$ probability and normalized by the mean and standard deviation for each color channel. We train the model via stochastic gradient decent (SGD) with momentum 0.9 and weight decay of $2 \times 10^{-4}$ for all experiments. All models are trained for 200 epochs setting mini-batch as 128 with same learning rate [4] for fair comparisons. The learning rate is set as [4]: the initial value is 0.1 and linear warm up at first 5 epochs. The learning rate is decayed at the $160^{th}$ and $180^{th}$ epoch by 0.01. For all *mixup* [31] and its extension methods [27, 7], we fine-tune the model after $120^{th}$ epoch. The $\alpha$ of Beta distribution is 0.5 for UniMix and 1.0 for others.

### C.2    Implement details on large-scale datasets

We adopt ResNet-10 & ResNet-50 for ImageNet-LT and ResNet-50 for iNaturalist 2018. For ImageNet-LT, ResNet-10 and ResNet-50 are trained from scratch for 90 epochs, setting mini-batch as $512$ and $64$, respectively. For iNaturalist 2018, we adopt the vanilla ResNet-50 with mini-batch as $512$ and train for 90 epochs. With proper data augment strategy [34], the SGD optimizer with momentum 0.9 and weight decay $1 \times 10^{-4}$ optimizes the model with same learning rate for fair comparisons [4, 34, 15]. We set base learning rate as 0.2 and decay it at the $30^{th}$ and $60^{th}$ epoch by 0.1, respectively.

## D    Additional Experiment Results

We present additional experiments for comprehensive comparisons with previous methods and make a detailed analysis, which can be summarized as follows:

• Visualized top-1 validation error (%) comparisons of previous methods.

- Additional quantity and visualized comparisons of classification *calibration*.

- Additional visualized comparisons of confusion matrix and $\log$-confusion matrix.

- Additional comparisons on test imbalance scenarios.

- Additional comparisons on state-of-the-art two-stage method.

## D.1 Visualized comparisons on CIFAR-10-LT and CIFAR-100-LT

Previous sections have shown the remarkable performance of the proposed UniMix and Bayias. Fig. D1 shows the visualized top-1 validation error rate (%) comparisons on CIFAR-10-LT and CIFAR-100-LT with $\rho \in \{10, 50, 100, 200\}$ for clear and comprehensive comparisons. The histogram indicates the value of each method. The positive error term represents its distance towards the best method, while the negative term indicates the advance towards the worst one.

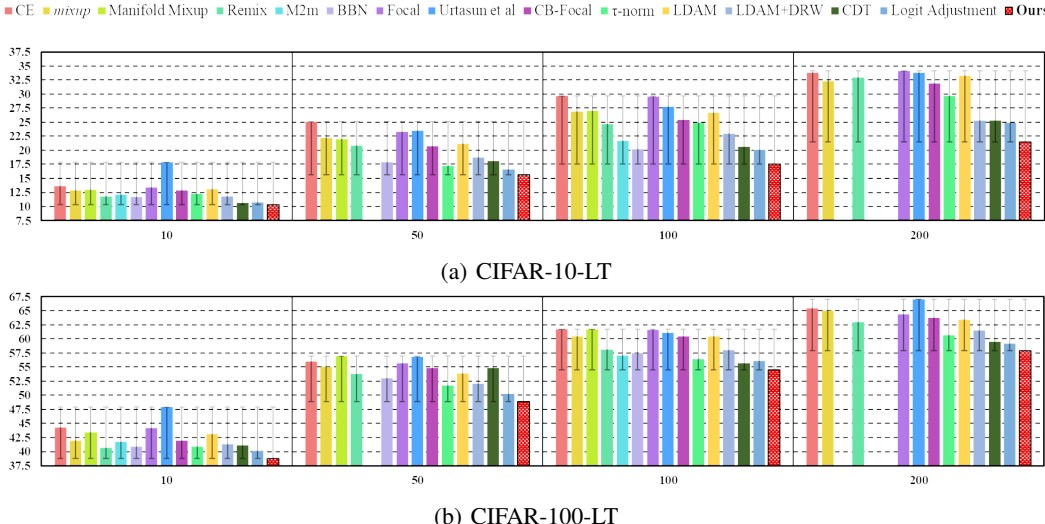

(a) CIFAR-10-LT

(b) CIFAR-100-LT

Figure D1: Visualized top1 error (%) ($y$-axis) comparisons of CIFAR-10-LT and CIFAR-100-LT on different $\rho$ ($x$-axis) in ResNet-32. The proposed method (red) achieves the lowest error rate compared with other methods. The leading advantage of our method increases consistently as imbalance factor gets larger.

Results in Fig. D1 show that the proposed method outperforms others with lower error rate over all imbalance factors settings. As the dataset gets more skewed and imbalanced, the advantage of our method gradually emerges. On the one hand, the proposed UniMix generates a tail-majority pseudo dataset favoring the tail feature learning, which makes the model achieve better *calibration*. It is practical to improve the generalization of all classes and avoid potential over-fitting and under-fitting risks. On the other hand, the proposed Bayias overcomes the bias caused by existing *prior* differences, which improves the model's performance on the balanced validation dataset. Even in extremely imbalanced scenarios (e.g., CIFAR-100-LT-200), the proposed method still achieves satisfactory performance.

## D.2 Additional quantitative and qualitative *calibration* comparisons

### D.2.1 Definition of *calibration*

*Calibration* means a model's predicting probability can estimate the representative of the actual correctness likelihood, which is vital in many real-world decision-making applications [23, 5, 2]. Suppose a dataset contains $N$ samples $\mathcal{D} := \{(x_k, y_k)\}_{k=1}^{N}$, where $x_i$ and $y_i$ represent the $i^{th}$ sample and its corresponding label, respectively. Let $\hat{p}_{y_i} = \mathbb{P}(Y = y_i | x = x_i)$ be the confidence of the predicted label, and divide dataset $\mathcal{D}$ into a mini-batch size $m$ with $M = N/m$ in total. The accuracy

(acc) and the confidence (cfd) in a mini-batch $\mathcal{B}_m$ are:

$$acc(\mathcal{B}_m) = \frac{1}{m} \sum_{i \in \mathcal{B}_m} \mathbb{1}(\hat{y}_i = y_i)$$
$$cfd(\mathcal{B}_m) = \frac{1}{m} \sum_{i \in \mathcal{B}_m} \hat{p}_{y_i} \tag{D.1}$$

According to Eq.D.1, a perfectly calibrated model will strictly have $acc(\mathcal{B}_m) \equiv cfd(\mathcal{B}_m)$ for all $m \in \{1, \cdots, M\}$. Hence, the Expected Calibration Error (ECE) is proposed as a scalar statistic of *calibration* to quantitatively measure classifiers' mean distance to the ideal $acc(\mathcal{B}_m) \equiv cfd(\mathcal{B}_m)$, which is defined as:

$$ECE = \sum_{m=1}^{M} \frac{|\mathcal{B}_m|}{n} |acc(\mathcal{B}_m) - cfd(\mathcal{B}_m)| \tag{D.2}$$

In addition, reliable confidence measures are essential in high-risk applications. The Maximum Calibration Error (MCE) describes the worst-case deviation between confidence and accuracy, which can be defined as:

$$MCE = \max_{m \in \{1, \cdots, M\}} |acc(\mathcal{B}_m) - cfd(\mathcal{B}_m)| \tag{D.3}$$

According to Eq.D.2,D.3, a well-calibrated model should coverage ECE and MCE to 0, indicating that the prediction score reflects the actual accuracy *likelihood*. Under this circumstances, the calibrated model will show excellent robustness and generalization, especially in LT scenarios. Although the authors in [10] propose the temperature scaling as a post-hoc method to adjust the classifier, our motivation is to train a calibrated model end to end without loss of accuracy.

Previous work [1] points out that ECE scores suffer from several shortcomings. Hence, additional metrics [22, 1] are proposed to show more robustness, e.g., Adaptivity & Adaptive Calibration Error (ACE), Thresholding & Thresholded Adaptive Calibration Error (TACE), Static Calibration Error (SCE), and Brier Score (BS). The definition of the above metrics are as follows:

TACE disregards all predicted probabilities that are less than a certain threshold $\epsilon$. It adaptively chooses the bin locations to ensure each bin has the same instance numbers and estimates the miscalibration of probabilities across all classes in the prediction while the ECE only chooses the top-1 predicted class. TACE is defined as Eq.D.4:

$$TACE = \frac{1}{CR} \sum_{c=1}^{C} \sum_{r=1}^{R} |acc(\mathcal{B}_r, c) - cfd(\mathcal{B}_r, c)| \tag{D.4}$$

where $acc(\mathcal{B}_r, c)$ and $cfd(\mathcal{B}_r, c)$ are the accuracy and confidence of adaptive calibration range $r$ for class label $c$, respectively. Calibration range $r$ defined by the $\lfloor N/R \rfloor_{th}$ index of the sorted and thresholded predictions, exclude the prediction less than $\epsilon$. If set $\epsilon = 0$, TACE converts to ACE.

SCE is a simple extension of ECE to every probability in the multi-class setting. SCE bins predictions separately for each class probability, computes the calibration error within the bin, and averages across bins. SCE is defined as Eq.D.5:

$$SCE = \frac{1}{C} \sum_{c=1}^{C} \sum_{b=1}^{B} \frac{n_{bc}}{N} |acc(\mathcal{B}_b, c) - cfd(\mathcal{B}_b, c)| \tag{D.5}$$

where $acc(\mathcal{B}_b, c)$ and $cfd(\mathcal{B}_b, c)$ are the accuracy and confidence of bin $\mathcal{B}_b$ for class label $c$, respectively. $n_{bc}$ is the prediction number in bin $\mathcal{B}_b$ for class $c$. $N$ is the total number of test set.

BS [3] has also been known as a metric for the verification of predicted probabilities. Similarly to the log-likelihood, BS penalizes low probabilities assigned to correct predictions and high probabilities assigned to wrong ones, which is defined as Eq.D.6:

$$BS = \frac{1}{NC} \sum_{i=1}^{N} \sum_{c=1}^{C} (\mathbb{1}(y_i^* = c) - \mathbb{P}(Y = y_c | X = x_i))^2 \tag{D.6}$$

where $N$ represents the total number of test set. $y_i^*$, $\mathbb{P}(Y = y_c | X = x_i)$ represent the ground truth and predicted label, respectively.

### D.2.2  Quantitative results of *calibration* on CIFAR-LT

Besides ECE and MCE results, we additionally provide quantitative metrics results, i.e., ACE, TACE setting threshold $\epsilon = 1e - 3$, SCE, and BS. The comparisons are illustrated in Tab.D1.

Table D1: Quantitative calibration metric of ResNet-32 on CIFAR-10/100-LT-100 test set. Smaller ACE, TACE, SCE, and BS indicate better calibration results. Either of the proposed methods achieves a well-calibrated model compared with others. The combination of UniMix and Bayias still achieves the best performance.

| Dataset | CIFAR-10-LT-100 | | | | CIFAR-100-LT-100 | | | |
|---|---|---|---|---|---|---|---|---|
| Calibration Metric ($\times 100$) | ACE | TACE | SCE | BS | ACE | TACE | SCE | BS |
| ERM | 5.42 | 4.66 | 5.46 | 53.61 | 0.831 | 0.709 | 0.952 | 97.05 |
| *mixup* [31] | 4.87 | 4.20 | 4.92 | 49.09 | 0.751 | 0.683 | 0.849 | 89.26 |
| Remix [7] | 3.49 | 3.32 | 3.77 | 44.84 | 0.740 | 0.675 | 0.846 | 89.33 |
| LDAM+DRW [4] | 4.14 | 2.84 | 4.43 | 44.76 | 0.801 | 0.671 | 1.093 | 107.4 |
| UniMix (ours) | 3.48 | 3.27 | 3.57 | 38.98 | 0.505 | 0.555 | 0.687 | 80.70 |
| Bayias (ours) | 2.45 | 2.40 | 2.69 | 34.30 | 0.460 | 0.518 | 0.631 | 78.78 |
| UniMix+Bayias (ours) | **2.31** | **2.17** | **2.43** | **32.84** | **0.450** | **0.517** | **0.623** | **78.24** |

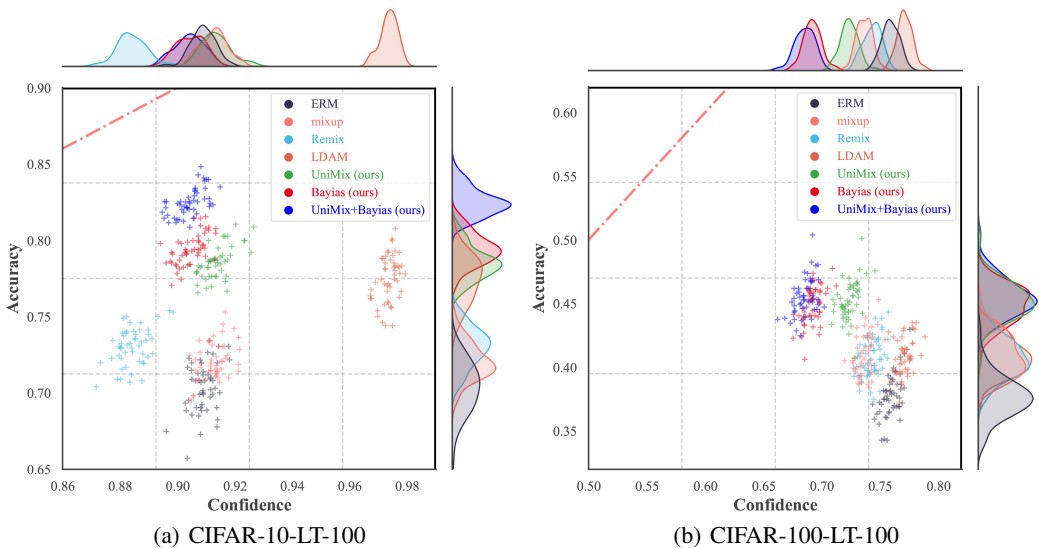

(a) CIFAR-10-LT-100          (b) CIFAR-100-LT-100

Figure D2: Additional comparisons of joint density plots of accuracy vs. confidence on CIFAR-10-LT and CIFAR-100-LT. A well-calibrated classifier's density will lay around the $y = x$ (red dot line). The combination of UniMix and Bayias achieves remarkable results especially in the severely imbalance scenarios (i.e., $\rho = 100$).

#### D.2.3 Additional visualized *calibration* comparison on CIFAR-LT

To make intuitive comparisons, we visualize additional confidence-accuracy joint density plots and reliability diagrams in CIFAR-LT test set setting $\rho = 100$. The confidence data is obtained by the average *Softmax* winning score in a test mini-batch [26]. The reliability diagrams data is obtained follow [10], which groups all prediction score into 15 interval bins and calculate the accuracy of each bin. The results are available in Fig.D2.

Fig.D2 clearly shows that the proposed method achieves better *calibration* in the CIFAR-LT test set. Each scatters data is the corresponding result of confidence and accuracy in a mini-batch $m$. In LT scenarios, the validation accuracy is usually lower than train accuracy, especially for the tail classes. Such overconfidence and miscalibration obstruct the network from having better generalization performance. In Fig.D2, the distribution of scattered points reflects a model's generalization and *calibration*. Our proposed method is the closest towards ideal $y = x$. It means each class gets enough regulation and hence contributes to a well-calibrated model. In contrast, *mixup* and its extensions are similar to ERM, which show limited improvement on classification *calibration* in LT scenarios. LDAM ameliorates the LT situation to some extent but results in more severe overconfidence cases, which may explain why it is counterproductive to other methods.

The reliability diagram is another visual representation of model *calibration*. As Fig.D3 shows, the baseline (ERM) is overconfident in its predictions and the accuracy is generally below ideal $y = x$

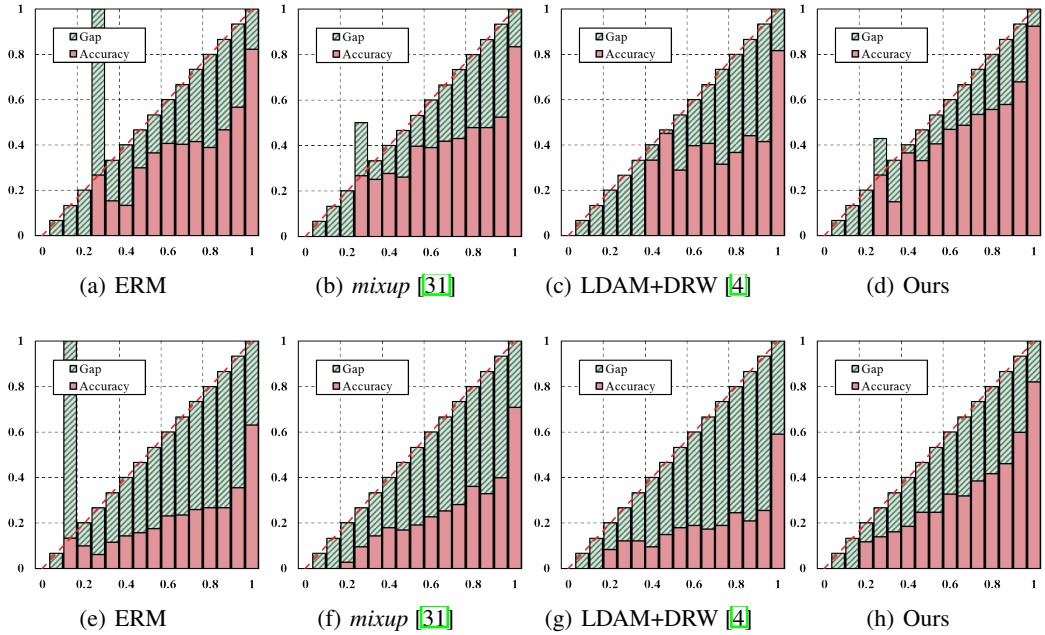

Figure D3: Reliability diagrams of ResNet-32 on CIFAR-10-LT-100 (top) and CIFAR-100-LT-100 (bottom). $x$-axis and $y$-axis represent the confidence and accuracy, respectively. A well-calibrated model shows smaller gap in each bin, i.e., the accuracy bar is closed to $y = x$. Our proposed method shows best results compared with others without any further fine-tune.

for each bin. Our method produces much better confidence estimates than others, which means our success in regulating all classes. Although some post-hoc adjustment methods [10, 33] can achieve better classification *calibration*, our approach trains a calibrated model end-to-end, which avoids the potential adverse effects on the original task. Considering the similarity of purposes with MiSLAS [33], we make detailed discussion in Appendix D.5.

## D.3 Additional visualized comparisons of confusion matrix on CIFAR-LT

### D.3.1 Visualized confusion matrix on CIFAR-10-LT

We further give additional visualized results on CIFAR-10-LT setting $\rho = 10$ and $\rho = 100$, which are available in Fig. D4, D5. The results show that all methods perform well compared with ERM in the simple dataset (e.g., CIFAR-10-LT-10, CIFAR-10-LT-100). In general, the proposed method significantly improves the accuracy of the tail for the larger diagonal values. The improvement on the tail is superior than other methods, which leads to state-of-the-art performance.

As for misclassification results, the non-diagonal elements concentrate on the top-right triangle. Hence, most of previous methods have limited improvement on minority classes, which tend to simply predict the instances in tail as majority ones. In contrast, the proposed method improves the performance for the tail and makes the misclassification case more balance distributed.

### D.3.2 Visualized $\log$-confusion matrix on CIFAR-100-LT

Furthermore, we visualize the distinguishing results of our methods on challenging CIFAR-100-LT for comprehensive comparisons. Specifically, we plot additional comparisons on CIFAR-100-LT-10 (Fig. D6) and CIFAR-100-LT-100 (Fig. D7).

Fig. D6, D7 show that in the challenging LT scenarios, the diagonal values get decreased towards the tail. A proper confusion matrix should exhibit a balanced distribution of misclassification case and large enough diagonal values. The proposed method outperforms others both in the correct cases (the diagonal elements) and the misclassification case distribution. In general, the proposed method shows the best performance, especially in the tail classes (deeper color represents larger value). The

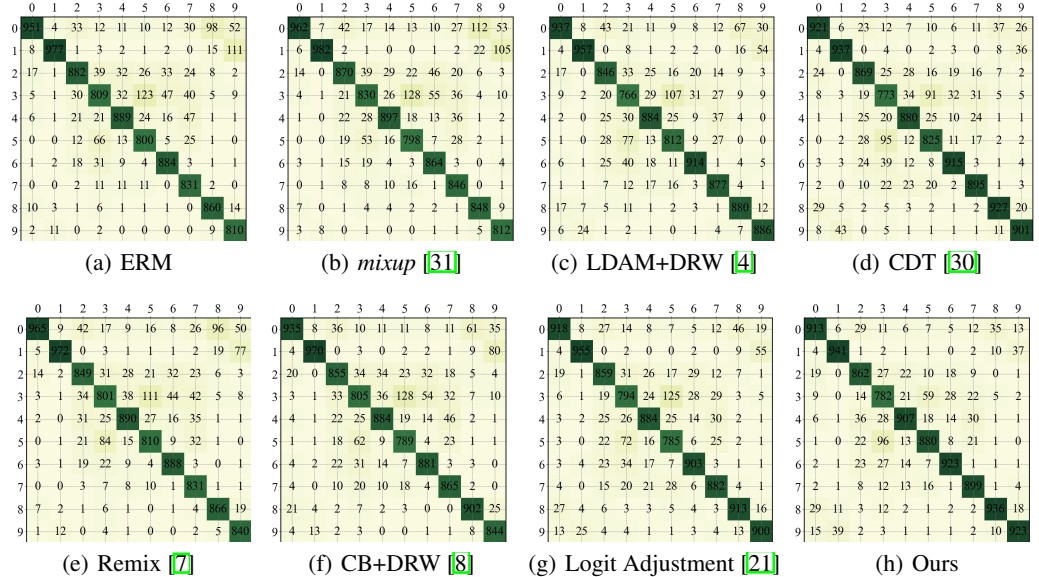

Figure D4: Additional confusion matrix comparisons on CIFAR-10-LT-10. The $x$-axis and $y$-axis indicate the ground truth and predicted labels, respectively. Deeper color indicates larger values. All methods achieve satisfactory results while ours further improves the tail feature learning and make misclassification cases more balanced distributed.

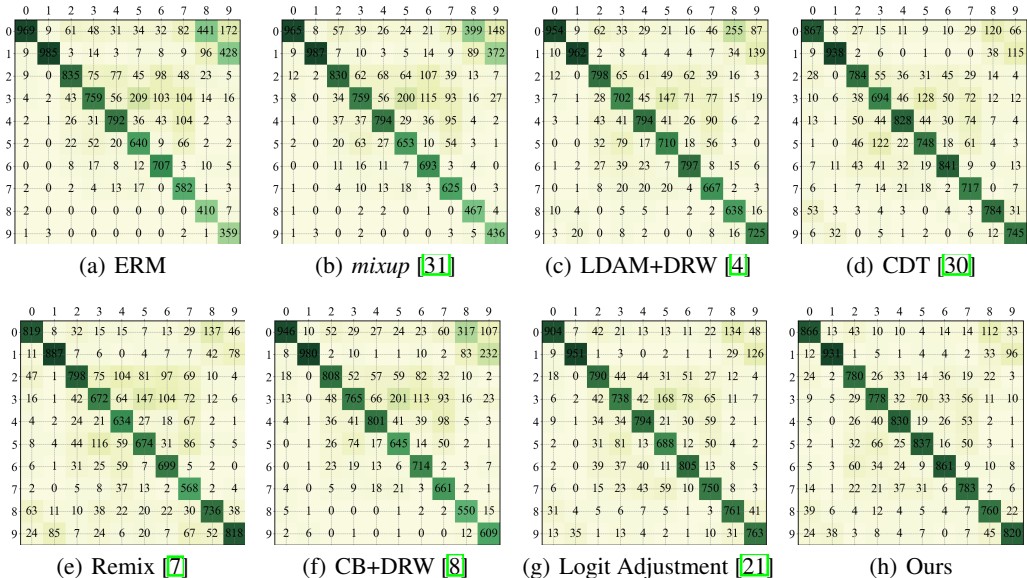

Figure D5: Additional confusion matrix comparisons on CIFAR-10-LT-100. The $x$-axis and $y$-axis indicate the ground truth and predicted labels, respectively. Deeper color indicates larger values. The disparity of these methods gradually gets appeared. Our method achieves the best accuracy as well as better non-diagonal distribution. Other methods exhibit an obvious bias towards the head or tail.

misclassification cases in other methods apparently concentrate on the upper triangular, which means the classifiers simply predict the tail samples as the head and vise versa.

## D.4 Results on test imbalance scenarios

Zhang et al. [32] have discussed the existing challenging and severe bias issue in imbalanced distributed test set. When the test set distribution is ideally known or well estimated, Bayias enables to overcome the bias issues in such scenarios by considering the priors. In contrast, CE loss implicitly

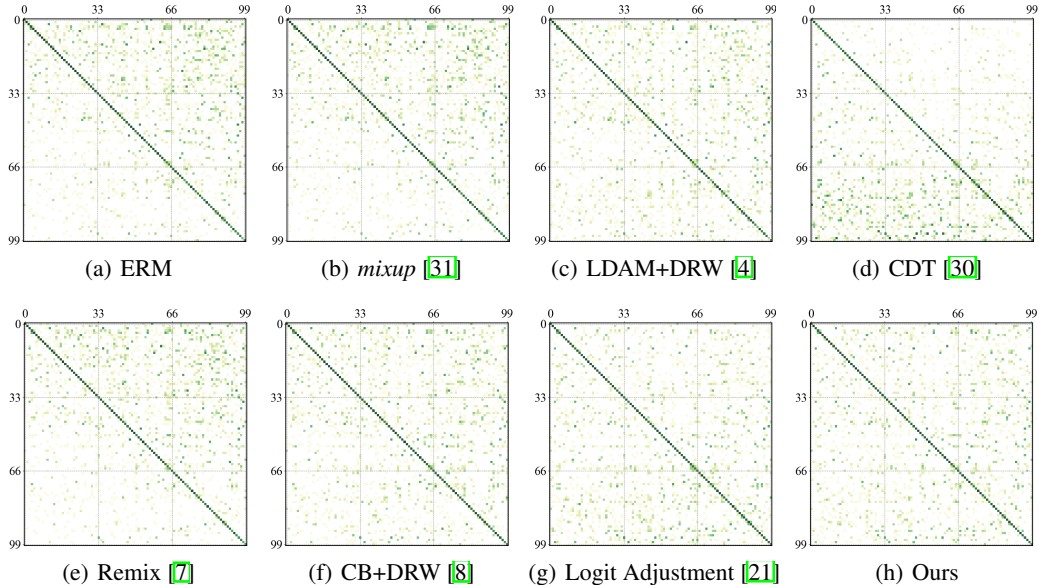

Figure D6: Additional $\log$-confusion matrix comparisons on CIFAR-100-LT-10. The $x$-axis and $y$-axis indicate the ground truth and predicted labels, respectively. $\log$ operation is adopted for clearer visualisation. Deeper color indicates larger values. The increased class number makes the misclassification distribution more clearly.

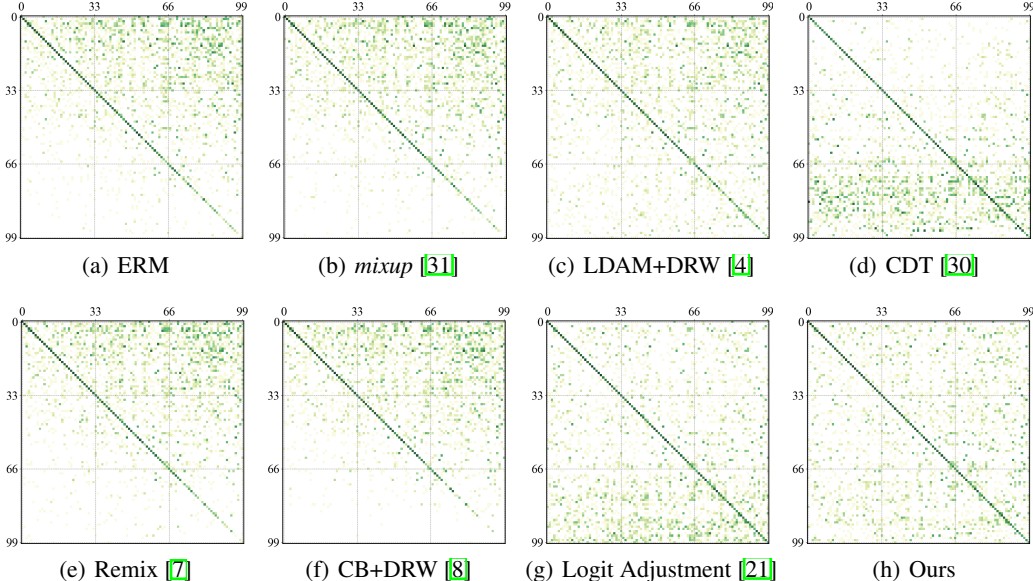

Figure D7: Additional $\log$-confusion matrix comparisons on CIFAR-100-LT-100. The $x$-axis and $y$-axis indicate the ground truth and predicted labels, respectively. $\log$ operation is adopted for clearer comparisons. Deeper color indicates larger values. Other methods show obviously poor generalization on the tail in the challenging CIFAR-100-LT-100, while the proposed method achieves significantly superior correct classification results and balanced misclassification case distribution.

requires train set and test set same distributed, while LA loss requires test set balanced distributed. LA and CE losses are powerless in other scenarios, while Bayias is flexible and capable with imbalanced distributed test sets, i.e., the constant term $\log C$ can change to be the test set distribution $\log \pi'$ to deal with such scenarios.

We sample test sets under different distribution with test imbalance factor $\rho'$ (i.e., long-tail ($\rho' > 1$), balanced ($\rho' = 1$), and reversed long-tail ($\rho' < 1$)), corresponding to different imbalance factor

100, 10, 1, 0.1, 0.01. The following comparisons in Tab.D2 show the necessity of considering test distribution's influence, which reveals our motivation to deal with bias issues. When train and test sets follow the same distribution, Bayias is equal to CE and outperforms LA. When the distribution of train and test set is different severely (e.g., $\rho' = 100$ and 0.01), Bayias outperforms both CE and LA. Similar experiments and conclusions can be obtained from [13] as well.

Table D2: Comparisons of CE, LA and Bayias loss on CIFAR-100-LT class-imbalanced test set. Note that the imbalanced test set is sampled by discarding some data. Therefore, the instance numbers of test samples vary in different imbalance factors, and longitudinal comparisons can reflect the effectiveness of Bayias.

| Train set | CIFAR-100-LT-10 | | | | | CIFAR-100-LT-100 | | | | |
|---|---|---|---|---|---|---|---|---|---|---|
| $\rho'$ | 100 | 10 | 1 | 0.1 | 0.01 | 100 | 10 | 1 | 0.1 | 0.01 |
| CE | 69.73 | 63.67 | 55.70 | 48.48 | 43.13 | 63.54 | 52.74 | 38.32 | 25.00 | 14.03 |
| LA | 61.94 | 60.14 | 59.87 | 56.12 | 54.06 | 59.13 | 51.06 | 43.89 | 31.14 | 26.42 |
| Bayias | **71.89** | **63.36** | **60.02** | **58.20** | **62.32** | **64.90** | **54.15** | **43.92** | **36.20** | **35.42** |

### D.5 Comparisons with the two-stage method

We propose UniMix and Bayias to improve model *calibration* by tackling the bias issues caused by imbalanced distributed train and test set. We improve the model calibration and accuracy simultaneously in an end-to-end manner. However, there are some methods to ameliorate calibration in post-hoc (e.g., temperature scaling [10]) or two-stage (e.g., label aware smoothing [33]) way. As we discussed above, such pipelines will be effective and we integrate our proposed methods with one of them for further comparison. The compared results are illustrated in Tab.D3.

Table D3: Comparisons with state-of-the-art two-stage method on CIFAR-100-LT.

| Dataset | | CIFAR-100-LT-10 | | CIFAR-100-LT-50 | | CIFAR-100-LT-100 | |
|---|---|---|---|---|---|---|---|
| Metric (%) | | Accuracy | ECE | Accuracy | ECE | Accuracy | ECE |
| MiSLAS | 1st-stage | 58.7 | 3.91 | 45.6 | 6.00 | 40.3 | 10.77 |
| | 2nd-stage | **63.2** | **1.73** | 52.3 | **2.25** | 47.0 | 4.83 |
| Ours | 1st-stage | 61.1 | 7.79 | 51.8 | 4.91 | 47.6 | 3.24 |
| | 2nd-stage | 63.0 | 1.98 | **52.6** | 3.45 | **48.3** | **1.44** |

MiSLAS adopts *mixup* for the 1st-stage and label aware smoothing (LAS) for 2nd-stage. We add the LAS in additional 10 epochs following our vanilla pipeline to build a two-stage manner. Our method generally outperforms MiSLAS in the 1st-stage and can achieve on par with accuracy and lower ECE. In specific, MiSLAS mainly improves accuracy and ECE in the 2nd-stage, i.e., classifier learning, by a large margin, which indicates LAS's effectiveness. We adopt the LAS to replace Bayias compensated CE loss in the 2nd-stage and can further improve accuracy and ECE, especially in the more challenging CIFAR-100-LT-100. However, the improvement of the 2nd-stage is not significant compared to MiSLAS. Since the LAS and Bayias compensated CE loss are both modifications on standard CE loss, it is hard to implement both of them simultaneously. We suggest that the LAS plays similar roles as Bayias compensated CE loss does, which explains the limited performance improvement.

## E Additional discussion

**What is the difference between UniMix and traditional over-sample approaches?** We take representative over-sample approach SMOTE [6] for comparisons. SMOTE is a classic over-sample method for imbalanced data learning, which constructs synthetic data via interpolation among a sample's neighbors in the same class. However, the authors in [26] illustrate that such interpolation without labels is negative for classification calibration. In contrast, *mixup* manners take elaborate-designed label mix strategies. The success of label smoothing trick [12] and *mixup*s imply that the fusion of label may play an important role in classification accuracy and calibration. Different from other methods, our UniMix pays more attention to the tail when mixing the label and thus achieves remarkable performance gain.

**Why choose $\xi^*_{i,j} \sim \mathscr{U}(\pi_{y_i}, \pi_{y_j}, \alpha, \alpha)$?** As we have discussed the motivation of UniMix Factor before, it is intuitive to set $\xi_{i,j} = \pi_{y_j}/(\pi_{y_i} + \pi_{y_j})$. Although satisfactory performance gain is obtained on CIFAR-LT in this manner, we fail on large-scaled dataset like ImageNet-LT. We suspect that the $\xi^*_{i,j}$ will be close to 0 or 1 when the datasets become extremely imbalanced and hence the effect of our mixing manner disappears. To improve the robustness and generalization, we transform the origin $Beta(\alpha, \alpha)$ distribution ($\alpha \leq 1$) to maximize the probability of $\xi_{i,j} = \pi_{y_j}/(\pi_{y_i} + \pi_{y_j})$ and its vicinity. We set $\alpha = 0.5$ on CIFAR-LT and keep the same $\alpha$ as *mixup* and Remix on ImageNet-LT and iNaturalist 2018.

**Why does Bayias work?** According to Bayesian theory, the estimated *likelihood* is positive correlation to *posterior* both in balance-distributed train and test set, with the same constant coefficient of *prior* and evidence factor. However, though evidence factor is regraded as a constant in LT scenarios, the LT dataset suffers seriously skewed *prior* of each category in the train set, which is extremely different from the balanced test set. Hence, the model based on maximizing *posterior* probability needs the to compensate the different priors firstly. From the perspective of optimization, Deep Neural Network (DNN) is a non-convex model, minuscule bias will eventually cause disturbances in the optimization direction, resulting in parameters converging to another local optimal. In addition, $\log(\pi_i) + \log(C)$ is consistent with traditional balanced datasets (i.e., $\pi_i \equiv 1/C$), which turns to be a special case (i.e., $\log(1/C) + \log(C) \equiv 0$) of Bayias. Furthermore, our Bayias can compensate any discrepancy between train set and test set via directly setting the Bayias as $\log(\pi_{y_i}) - \log(\pi'_{y_i})$, where $\pi_{y_i}$ is the label prior in train set and $\pi'_{y_i}$ is the label prior in test set.