# OpenReview forum: "Towards Calibrated Model for Long-Tailed Visual Recognition from Prior Perspective"
_NeurIPS.cc/2021/Conference — NeurIPS 2021 Poster_

### Official Review · Reviewer_V6xE · 2021-06-27

**Rating:** 7
**Confidence:** 5

**Summary:**

This paper studies a practical problem, namely long-tailed recognition. To address this problem, this paper proposes two methods: UniMix and compensated cross-entropy loss. To be specific, UniMix adopts a balanced data sampler to improve the number of head-tail pairs for mixup. Experimental results show the effectiveness of the proposed method.



**Limitations And Societal Impact:**

Please refer to the main review for the concerns of this paper. I would like to see the response from the authors.

**Main Review:**

Pros:
1. This paper reveals an interesting problem in existing mixup-based long-tailed recognition (LT) methods, i.e., they will generate more head-head pairs since there exist more samples from the head classes. Such a problem may lead to more imbalanced learning.
2. To address the above problem in existing mixup-based LT methods, this paper proposes an intuitive UniMix strategy.
3. The proposed methods are theoretically analyzed.
4. Extensive results show the effectiveness of the proposed method.

Cons:
1. When this paper analyzes the bias issue of mixup, it assumes the instance number is exponential with $\lambda$. I wonder whether the result is the same when the instance number is not exponential with $\lambda$, since real-world datasets may not follow such a way.
2. For the proposed compensated cross-entropy loss, it is very similar to the balance softmax [1], which is also obtained from the Bayesian inference's perspective. Moreover, similar ideas of logit adjustment and rebalanced mixup have been deeply explored in the work [2,3]. Although this is not the drawback of this paper, I do not think this loss can be regarded as an important contribution.

    [1] Balanced Meta-Softmax for Long-Tailed Visual Recognition, NeurIPS 2020.

    [2] Long-tail Learning Via Logit Adjustment, ICLR 2021.

    [3] Remix: Rebalanced Mixup. arXiv, 2020.

3. In the final algorithm, there are actually two phases, i.e., first mixup training for T1 steps, and then non-mixup training for T2-T1 steps. I understand this is a trick to improve performance [4], but I wonder how effective the first phase is, since the main contribution of this paper is at phase 1, i.e., UniMix.

    [4] Bag of Tricks for Long-Tailed Visual Recognition with Deep Convolutional Neural Networks, AAAI 2021.

4. As proposed in this paper, mixup leads to bias distribution (since generating more head-head pairs) and has a negligible contribution to model calibration. Regarding this, why do existing mixup-based methods also obtain very promising performance?  Does this mean that model calibration may not be an essential metric for the long-tailed recognition problem?

5. Following question 4, according to Tables 3 and D1, LDAM is harmful to model calibration. However, LDAM is also effective and helps handle the imbalance issue. Does this mean that a method with worse model calibration does not essentially mean a bad long-tailed method, and a method with better model calibration does also not mean a better long-tailed method? These two concerns are important since this paper is based on model calibration.

6. Why not compare to MiSLAS [5]. This paper analyzes mixup for model calibration and thus is very related to the submission. One interesting thing is that from the results reported in two papers, although MiSLAS only uses Mixup and ignores the head-head pair bias, it obtain even better accuracy and lower ECE than the method in this submission. I understand there may exist some differences in methodology (1-stage method v.s. 2-stage method), but it would be better for authors to also discuss MiSLAS and compared with it.
In addition, one interesting thing is that MiSLAS finds that mixup is beneficial to representation learning and bad for classifier learning. From this perspective, the head-head bias seems to mainly influence classifier learning. Could you discuss this？

    [5] Improving Calibration for Long-Tailed Recognition, CVPR 2021.

***

****Post Rebuttal****

The most important contribution of this paper is that it theoretically and empirically reveals an interesting problem in existing mixup-based long-tailed recognition (LT) methods, i.e., they will generate more head-head pairs that may lead to more imbalanced learning.

To address the argued problem, this paper proposes a new method, consisting of (1) a Unimix strategy for generating more tail pairs and (2) a Bayias loss for rebalancing the optimization. Nevertheless, the performance improvement of Unimix over mixup is limited, while most performance improvement is derived from the Bayias loss (c.f. Table 3 in the submission and the new results in the response). Moreover, the novelty of the Bayias loss is limited, considering the logits adjustment loss [1] and balanced softmax loss [2]. For the balanced test set, these losses perform similarly. Although the authors argue that the extension of Bayias loss with $\log(C)$ to $\log(\pi')$ can help to handle imbalanced test class distribution when the test class distribution $\pi'$ is known [3], I think that it is very difficult to obtain the test class distribution in real-world applications [4], so the practicability and contribution of the extension are limited.

Considering the above advantages and limitations, the paper seems really a borderline paper and is difficult to decide. Therefore, I return to consider the value and contribution of this paper to the LT community. In my view, although the performance and novelty of the proposed method are not that promising, this paper theoretically reveals an important head-head bias problem in the existing mixup-based LT method, which is insightful to the LT community and even inspires more augmentation-based LT methods. **Considering that performance-only judgment should not be encouraged in the current AI community and novelty can be varied, I believe the contribution of this paper is good enough and decide to increase my score to 7**. Even so, I encourage the authors to further improve the proposed method for better performance, keep exploring augmentation-based LT methods, and try to solve more practical test-agnostic LT task in the future. Thanks again for your effort.

[1] Long-tail Learning Via Logit Adjustment, ICLR 2021.

[2] Balanced Meta-Softmax for Long-Tailed Visual Recognition, NeurIPS 2020.

[3] Disentangling Label Distribution for Long-tailed Visual Recognition. CVPR, 2021.

[4] Test-Agnostic Long-Tailed Recognition by Test-Time Aggregating Diverse Experts with Self-Supervision. ArXiv, 2021.

**Time Spent Reviewing:**

10+

---

> ### Author Response · Authors · 2021-08-10
> **Author Response to Reviewer V6xE**
>
> We sincerely thank Reviewer V6xE for the insightful and positive feedback. Our responses to the reviewer’s questions and concerns are as follows:
>
> *\# 1. When this paper analyzes the bias issue of mixup, it assumes the instance number is exponential with $\lambda$. I wonder whether the result is the same when the instance number is not exponential with $\lambda$, since real-world datasets may not follow such a way.*
> --------------------------------------------------------------------------------------------------------------------------------------------------------------------------------------------------------------------------------------------------------------------------------
>
> The conclusion of mixup is the same. The reasonable assumption of exponential distribution parameterized by $\lambda$ is followed [1], which is common in most works. It enables us to obtain the analytical solution results shown in this paper. Actually, we don't enforce $P\sim E(\lambda)$, according to the proves in Appendix A.3, for any distribution of $P$, we can still draw the same conclusion that the mixup $D_\nu$ follows the same distribution as $D$.
>
> [1]. Class-balanced loss based on effective number of samples, CVPR 2019
>
> *\# 2. The reason why we regard Bayias as an important contribution.*
> -----------------------------------------------------------------
>
> Thanks for raising this concern. We argue that Bayias is an essential contribution, and we'd highlight the reason for our point as follows:
>
> As discussed in Appendix B3, we analyze the difference between Bayias and other loss modification methods. Specifically, the Bayias-compensated CE loss has an additional constant term $\log (C)$ to deal with the bias issue of imbalanced data when the train set is LT and the test set is balanced.
>
> Bayias is a modification and extension of logit adjustment and Balanced Meta-Softmax. From the official code implement, we find that Balanced Meta-Softmax is very similar to LA loss. Hence, we discuss the difference between LA and Bayias here:
>
> - Note that LA loss has a scale factor which is a hyper-parameter to adjust the margin influence (LA is equal to our Bayias iff scale=1 and without $\log (C)$ term). LA makes an ablation study of scale factor (Fig.3 in [2]) in post-hoc setting, and the best value of this scale is different for different datasets. However, according to our proves in Appendix B, the scale factor should be identically equal to 1, and $\log (C)$ should be compensated (while LA misses such term).
>
> - When the test set is not strictly balanced distributed (as the real world inference cases are), LA cannot deal with such a situation because they are deduced based on the balanced error rate. In contrast, Bayias is capable of such scenarios by changing the constant term $\log (C)$ into the real data distribution $\log (\pi')$.
>
> *\# 3. How effective the first phase is?*
> -------------------------------------------------------------------------------------------------------------------------------------------------------------------------------------------------------------------------------------------------------------------------------------------------------------------------------------
>
> Thanks for the question. We need to discuss the difference between our algorithm and other 2-stage methods first. The main misunderstanding is that we remove UniMix in the last epochs. While 2-stage methods are: disentangle feature learning and classifier learning, the most obvious difference is that these methods (e.g., cRT, MiSLAS) will freeze the feature network or change loss function in the 2nd-stage to fine-tune. In contrast, our 1-stage method optimizes feature extractor and classifier simultaneously.
>
> We have shown the effectiveness of UniMix alone in Fig.1, Tab.3, and Appendix D.2. The results show that UniMix outperforms other mixup methods in accuracy and calibration and can further improve performance when integrating with other loss modification methods.

---

> > ### Author Response · Authors · 2021-08-10
> > **Author Response to Reviewer V6xE (continue)**
> >
> > *\# 4. Why do existing mixup-based methods also obtain very promising performance? Does this mean that model calibration may not be an essential metric for the long-tailed recognition problem?*
> > -----------------------------------------------------------------------------------------------------------------------------------------------------------------------------------------------------------------------------------------------------------------------------------------------------------------------------------------------------------------------------
> >
> > According to the results in confusion matrices, calibration is an essential metric to show the robustness in LT scenarios. A well-calibrated model will show proper generalization of the tail classes.
> >
> > [3] analyzes the reason why mixup is effective in balanced datasets. However, mixup is just passable in LT scenarios (see Tab.3).
> >
> > We agree that mixup is helpful for feature learning, which is the reason for its improvement of accuracy. Although mixup is negligible in the tail as we analyzed in Sec.2, mixup still causes more head-head pairs thus it can improve the head feature learning. The confusion matrices (see Fig.D4-D7) also show mixup's improvement in head samples. However, it has little help in the tail, which limits its contribution to calibration. This is exactly UniMix aims at solving.
> >
> > [3]. On Mixup Training: Improved Calibration and Predictive Uncertainty for Deep Neural Networks, NeurIPS 2019
> >
> > *\# 5. Does this mean that a method with worse model calibration does not essentially mean a bad long-tailed method, and a method with better model calibration does also not mean a better long-tailed method?*
> > -----------------------------------------------------------------------------------------------------------------------------------------------------------------------------------------------------------------------------------------------------------------------------------------------------------------------------------------------------------------------------------------------------------------------------------------------------------------
> >
> > That's a good question. Actually, the metric of methods is multiple. Apart from accuracy, calibration is also essential. Specifically, one model may achieve satisfactory accuracy (e.g., LDAM) but lacks calibration (see Appendix D1).
> >
> > According to our experiments, accuracy and calibration are not necessary and sufficient condition. However, empirical study in Tab.3 shows that calibration ensured methods combined tend to improve the model accuracy, while the poor calibrated methods tend to be counterproductive on accuracy when combined with other methods.
> >
> > Noteworthy, we'd explain the reason for LDAM's poor calibration here. Since LDAM is motivated by face loss (e.g., arcloss), it preserves weight norm operation, which leads to the output logits become very small. Therefore, though each logit is multiplied with a constant scale factor provide by [4] during training, LDAM still proves to be poor-calibrated, i.e., the confidence cannot correctly reflect the actual accuracy.
> >
> > We emphasize that a good accuracy model isn't bound to be well-calibrated, i.e., it isn't bound to be a well LT model. However, a well-calibrated model shows satisfactory generalization overall classes, which is significant in the model's deployment.
> >
> > The empirical study shows that although methods with worse model calibration do not essentially mean a bad LT method in terms of accuracy, those poor-calibrated methods are harmful combined with each other, which leads to counterproductive results on accuracy. On the contrary, a calibration-ensured method will show better generalization, and their combination tends to improve the accuracy further.
> >
> > [4]. Learning imbalanced datasets with label-distribution-aware margin loss, NeurIPS 2019
> >
> > *\# 6. Discuss of MiSLAS*
> > -------------------------
> >
> > Thanks for your suggestions. MiSLAS is published near submission, so we don't compare it in the current submission. We will add detailed discussion in the final version.
> >
> > We analyze MiSLAS that integrates Label Aware smooth loss + Shift BN learning + cRT training manner, which is actually a 2-stage method that disentangles representation learning and classifier learning. The baseline cRT is far better than our baseline ERM. The reason MiSLAS outperforms others lies in such 2-stage approach and cRT baseline, not merely mixup, and the train epoch of MiSLAS is **200+10**, while ours are **200**. Therefore, it is unfair to compare our methods that use ERM as the baseline in a 1-stage mode.
> >
> > When we adopt our methods into 2-stage approach as MiSLAS for fair comparisons (i.e., 2-stage vs. 2-stage， cRT vs. cRT, et al.), results are as follows. Our method outperforms MiSLAS in the first stage can achieve on par with accuracy and lower ECE. In specific, MiSLAS mainly improves the 2nd-stage, i.e., classifier learning. We adopt the improved MiSLAS in the 2nd-stage and can further improve accuracy and ECE. In the table, we use UniMix+Bayias on the 1st-stage (feature learning stage) and the methods of [5], i.e., label aware smooth and shift batch normalization learning in the 2nd-stage (fine-tune classifier stage).
> >
> > |  Dataset   |           | CIFAR-100-LT-10 |          | CIFAR-100-LT-50 |          | CIFAR-100-LT-100 |          |
> > | :--------: | --------- | :-------------: | :------: | :-------------: | :------: | :--------------: | :------: |
> > | Metric (%) |           |  **Accuracy**   | **ECE**  |  **Accuracy**   | **ECE**  |   **Accuracy**   | **ECE**  |
> > |   MiSLAS   | 1st-stage |      58.7       |   3.91   |      45.6       |   6.00   |       40.3       |  10.77   |
> > |            | 2nd-stage |    **63.2**     | **1.73** |      52.3       | **2.25** |       47.0       |   4.83   |
> > |    Ours    | 1st-stage |      61.1       |   7.79   |      51.8       |   4.91   |       47.6       |   3.24   |
> > |            | 2nd-stage |      63.0       |   1.98   |    **52.6**     |   3.45   |     **48.3**     | **1.44** |
> >
> > We agree the [5]'s point that mixup is beneficial to representation learning and bad for classifier learning. According to our proves, mixup generates imbalanced data that mainly contain head-head pairs, which is harmful to model calibration. Although mixup improves representation learning of head samples, it is incapable of dealing with bias issues (as we proved the existence of Bayias between imbalanced datasets), i.e., it is bad for classifier learning. When we use our proposed UniMix and Bayias to deal with bias issues in LT, such problems get relieved.
> >
> > Our additional experiment on MiSLAS finds that mixup strategies will contribute to calibration a lot in 2 stage manner, especially only adopt mixups in the 1st-stage, including our UniMix as well. However, the accuracy performance in the first stage will be unsatisfactory under this circumstance. Hence, the 2nd-stage for the classifier is necessary and crucial. From this observation, UniMix contributes to feature learning a lot. According to our experiment in the 2-stage approach, the evidence shows that our method is capable of achieving robust feature learning and achieves well results on both accuracy and ECE.
> >
> > [5]. Improving Calibration for Long-Tailed Recognition, CVPR 2021.

---

> > > ### Comment · Reviewer_V6xE · 2021-08-13
> > > **Thanks for response**
> > >
> > > Thanks very much for your response, which addresses most of my concerns. I think the discussion of "Calibration is not necessary and sufficient condition" is interesting.
> > > However, two concerns still exist.
> > > ***
> > >
> > > (1) In response of Question 2, I cannot directly see the value of $log(C)$. Can you provide **some empirical results?**
> > > In addition, you argue that "When the test set is not strictly balanced distributed (as the real world inference cases are), LA cannot deal with such a situation because they are deduced based on the balanced error rate. In contrast, Bayias is capable of such scenarios by changing the constant term $log(C)$ into the real data distribution $log(\pi')$". Although it is okay for me, the mentioned modified Bayias-compensated CE loss is quite similar to **the Post-compensated softmax loss in LADE [1].** In addition, I think it is better and more rigorous to change "When the test set is not strictly balanced distributed" to "When the test set is not strictly balanced distributed and the test class distribution is known". Since **when the test class distribution is unknown (i.e., test-agnostic LT[2]),** such a modification is unavailable and the strength is missing.
> > >
> > > [1] Disentangling Label Distribution for Long-tailed Visual Recognition. CVPR, 2021.
> > >
> > > [2] Test-Agnostic Long-Tailed Recognition by Test-Time Aggregating Diverse Experts with Self-Supervision. ArXiv, 2021.
> > >
> > > ***
> > > (2) In response of Question 3, I totally understand the difference between 1-stage and 2-stage methods. My question is that **when removing the non-mixup training,** how effective would be for UniMix after addressing head-head bias.
> > > ***
> > >
> > > The new results in the response of Question 6 show that two-stage UniMix performs comparably to MiSLAS. That is, UniMix mainly improves 1-stage LT methods, and the improvement of UniMix over standard mixup on representation learning would not be accumulated to improve the class-balanced classifier learning of 2-stage methods. Such a discussion would make the characteristic of UniMix more clear, which improves the quality of this paper and may inspire more researchers to think deeper. Expect to your further discussion on the above two remaining concerns.

---

> > > > ### Author Response · Authors · 2021-08-19
> > > > **Response to Reviewer V6xE**
> > > >
> > > > *\# 1. Some empirical results to explain the value of $\log(C)$ term.*
> > > > ------------------
> > > >
> > > > Thanks for the Reviewer V6xE's response. Bayias is equal to LA in terms of $\log (C)$ is a constant when the test set is balanced, so we'd make additional comparisons with the imbalanced test set to illustrate the value of $\log (\pi')$ term. The motivation of Bayias is to modify the bias caused by different label distribution datasets. Although we have an inconsistent starting point as LA, a similar result with additional margin term $\log (\pi')$ is deduced compared with LA. That's the reason we regard it as a complete form and extension of LA. We're encouraged that [1] also notice such bias issues in LT scenarios, and apologize for missing this paper before our submission. Compared with [1], [1] focused more on empirical calibration results and designed a new loss to deal with LT scenarios, while we are inspired by LA and deduce uniform pair-wise loss as LA to discuss further. We prove that Bayias ensures calibration and is suitable for different distributed test set theoretically and empirically.
> > > >
> > > > We agree that the strength of Bayias depends on the known test class distribution, so we make additional comparisons under controlled test class distribution here. $\log (C)$ term reveals the different distribution of train and test sets. Although the $\log (\pi')$ term plays a weak role in balanced distribution compared to LA, it is essential in other imbalanced test scenarios. We sample test sets under different distribution (i.e., long-tail($>1$), balanced($=1$), and reversed long-tail($<1$)), corresponding to different imbalance factor $100,10,1,0.1,0.01$. The following comparisons show the necessity of considering test distribution's influence, which reveals our motivation to deal with bias issues. Note that the imbalanced test set is sampled by discarding some data. Therefore, the instance numbers of test samples vary in different imbalance factors, and longitudinal comparisons can reflect the effectiveness of Bayias. **When train and test sets follow the same distribution, Bayias is equal to CE and outperforms LA. When the distribution of train and test set is different severely (e.g., $\rho$=100 and 0.01), Bayias outperforms both CE and LA.**
> > > >
> > > >
> > > > | Train set             |   CIFAR-100-LT-10  |       |                |                |                |  CIFAR-100-LT-100  |                |                |                |                |
> > > > |-----------------------|:--------------:|:-----:|:--------------:|:--------------:|:--------------:|:--------------:|:--------------:|:--------------:|:--------------:|:--------------:|
> > > > | Test imbalance factor |       100      |   10  |        1       |       0.1      |      0.01      |       100      |       10       |        1       |       0.1      |      0.01      |
> > > > | CE                    |      69.73     | **63.67** |      55.70     |      48.48     |      43.13     |      63.54     |      52.74     |      38.32     |      25.00     |      14.03     |
> > > > | LA                    |      61.94     | 60.14 |      59.87     |      56.12     |      54.06     |      59.13     |      51.06     |      43.89     |      31.14     |      26.42     |
> > > > | Bayias                | **71.89** | 63.36 | **60.02** | **58.20** | **62.32** | **64.90** | **54.15** | **43.92** | **36.20** | **35.42** |
> > > >
> > > > *\# 2. Comparsions with mixup, Remix, and UniMix without non-mixup training.*
> > > > ------------------
> > > >
> > > > Thanks for letting us know your concerns. Here we compare UniMix, mixup and Remix without non-mixup training for 200 epochs. To eliminate the impact of different code architectures on the results, we compare methods based on different repos, including 1) LDAM [3] (our code implement follows this work) and 2) MiSLAS [2] (we compare the 1st-stage) for fair comparisons. Although MiSLAS also follows LDAM, the official MiSLAS repo shows higher baseline (ERM) performance than the official LDAM repo, which can be seen in the following table.
> > > >
> > > > We split the test set equally by the number of instances into 3-parts as [2]: $H$: head, $M$: medium, $T$: tail (corresponding to $0-35,36-70,71-99$ classes), and give each part corresponding average accuracy. UniMix enables $tail$ classes to have better generalization and accuracy than others, which verifies the effectiveness of our method on dealing with head-head bias.
> > > >
> > > >
> > > > | Dataset            | CIFAR-100-LT-10 |                |                |                | CIFAR-100-LT-100 |                |               |                |
> > > > |--------------------|:---------------:|:--------------:|:--------------:|:--------------:|:----------------:|:--------------:|:-------------:|:--------------:|
> > > > | (LDAM et al. repo) |        H        |        M       |        T       |       All      |         H        |        M       |       T       |       All      |
> > > > |         ERM        |      68.25      |      56.34     |      39.48     |      55.70     |       64.42      |      36.51     |      8.03     |      38.32     |
> > > > |        mixup       |  **69.28** | **57.69** |      40.66     |      56.92     |       64.81      |      37.69     |      8.86     |      39.09     |
> > > > |        Remix       |      69.08      |      57.26     |      40.93     |      56.78     |       65.50      |      35.91     |      9.35     |      38.86     |
> > > > |       UniMix       |      60.83      |      57.51     | **53.45** | **57.53** |  **65.56**  | **38.77** | **9.62** | **39.96** |
> > > >
> > > >
> > > > | Dataset              | CIFAR-100-LT-10 |                |                |                | CIFAR-100-LT-100 |                |                |                |
> > > > |----------------------|:---------------:|:--------------:|:--------------:|:--------------:|:----------------:|:--------------:|:--------------:|:--------------:|
> > > > | (MiSLAS et al. repo) |        H        |        M       |        T       |       All      |         H        |        M       |        T       |       All      |
> > > > |          ERM         |      70.50      |      58.66     |      43.83     |      58.62     |       67.53      |      38.17     |      9.45      |      40.41     |
> > > > |         mixup        |      73.19      |      59.86     |      40.86     |      59.15     |  **73.56**  |      39.54     |      5.41      |      40.91     |
> > > > |         Remix        |  **73.56** | 60.51 |      40.79     |      59.49     |       69.50      |      40.06     |      6.45      |      40.49     |
> > > > |        UniMix        |      70.03      |     **61.31**     | **49.03** | **60.89** |       66.69      | **43.91** | **11.66** | **42.76** |
> > > >
> > > >
> > > > [1] Disentangling Label Distribution for Long-tailed Visual Recognition. CVPR, 2021.
> > > >
> > > > [2] Improving Calibration for Long-Tailed Recognition. CVPR, 2021.
> > > >
> > > > [3] Learning Imbalanced Datasets with Label-Distribution-Aware Margin Loss. NeurIPS, 2019.

---

> > > > > ### Comment · Reviewer_V6xE · 2021-08-20
> > > > > **Thanks for response**
> > > > >
> > > > > Thanks for the new results. I have one misunderstanding about the results without non-mixup training.
> > > > >
> > > > > The results in the table show that Unimix performs better than CE, Remix and Mixup. I think this can demonstrate the effectiveness of unimix. Following this, I  wonder how much the non-mixup training contributes to the performance, so I read Table 3 in the submission to compare UniMix (w/ non-mixup training) and UniMix (w/o non-mixup training) under the LDAM loss on CIFAR100-LT-100. However, I find that the performance of mixup+LDAM and remix+LDAM is inconsistent between Table 3 in the submission and the results in the response, as listed below:
> > > > >
> > > > > |  CIFAR-100-LT-100	   | Table 3 in submission  | Table in the response  |
> > > > > |  ----  | :----: | :----: |
> > > > > |  mixup+LDAM  |        40.22      |  	39.09    |
> > > > > |  Remix+LDAM |         40.59     |   	38.86 |
> > > > > |  UniMix (w/ non-mixup training)+LDAM  |        41.67   |     -  |
> > > > > |  UniMix (w/o non-mixup training)+LDAM  |       -   |   	39.96   	 |
> > > > >
> > > > > I wonder what factors lead to such inconsistency that I may ignore. After that, I will reconsider this paper and make my final decision. Thanks again for your hard work.

---

> > > > > > ### Author Response · Authors · 2021-08-20
> > > > > > **Response to Reviewer V6xE**
> > > > > >
> > > > > > Thanks for your reply. We are sorry that our expression misleads you. Actually, in the above response, the notions of **(LDAM et al. repo)** and **(MiSLAS et al. repo)** indicate that we conduct experiments based on these official repos, rather than mixup methods with LDAM or MiSLAS. All results are obtained **by only using mixup/Remix/UniMix w/o non-mixup training**. Our code implement follows LDAM, while the official MiSLAS repo shows higher baseline (ERM) performance than the official LDAM repo. Hence, we show two tables with different baselines to eliminate the impact of code architectures for fair comparisons.

---

> > > > > > > ### Comment · Reviewer_V6xE · 2021-08-20
> > > > > > > **Thanks for response**
> > > > > > >
> > > > > > > Thanks for the clarification. I summarize my final judgment below.
> > > > > > >
> > > > > > > I think the most important contribution of this paper is that it theoretically and empirically reveals an interesting problem in existing mixup-based long-tailed recognition (LT) methods, i.e., they will generate more head-head pairs that may lead to more imbalanced learning.
> > > > > > >
> > > > > > > To address the argued problem, this paper proposes a new method, consisting of (1) a Unimix strategy for generating more tail pairs and (2) a Bayias loss for rebalancing the optimization. Nevertheless, the performance improvement of Unimix over mixup is limited, while most performance improvement is derived from the Bayias loss (c.f. Table 3 in the submission and the new results in the response). Moreover, the novelty of the Bayias loss is limited, considering the logits adjustment loss [1] and balanced softmax loss [2]. For the balanced test set, these losses perform similarly. Although the authors argue that the extension of Bayias loss with $\log(C)$ to $\log(\pi')$ can help to handle imbalanced test class distribution when the test class distribution $\pi'$ is known [3], I think that it is very difficult to obtain the test class distribution in real-world applications [4], so the practicability and contribution of the extension are limited.
> > > > > > >
> > > > > > > Considering the above advantages and limitations, the paper seems really a borderline paper and is difficult to decide. Therefore, I return to consider the value and contribution of this paper to the LT community. In my view, although the performance and novelty of the proposed method are not that promising, this paper theoretically reveals an important head-head bias problem in the existing mixup-based LT method, which is insightful to the LT community and even inspires more augmentation-based LT methods. **Considering that performance-only judgment should not be encouraged in the current AI community and novelty can be varied, I believe the contribution of this paper is good enough and decide to increase my score to 7**. Even so, I encourage the authors to further improve the proposed method for better performance, keep exploring augmentation-based LT methods, and try to solve more practical test-agnostic LT task in the future. Thanks again for your effort.
> > > > > > >
> > > > > > > [1] Long-tail Learning Via Logit Adjustment, ICLR 2021.
> > > > > > >
> > > > > > > [2] Balanced Meta-Softmax for Long-Tailed Visual Recognition, NeurIPS 2020.
> > > > > > >
> > > > > > > [3] Disentangling Label Distribution for Long-tailed Visual Recognition. CVPR, 2021.
> > > > > > >
> > > > > > > [4] Test-Agnostic Long-Tailed Recognition by Test-Time Aggregating Diverse Experts with Self-Supervision. ArXiv, 2021.

---

### Official Review · Reviewer_1uZE · 2021-07-14

**Rating:** 4
**Confidence:** 4

**Summary:**

The authors propose two models to improve calibration for long-tailed recognition. 1) a Uniform Mixup is proposed, which adopts an advanced mixing factor and sampler in favor of the minority. 2) the authors propose a Bayias loss to compensate for the prior bias. The proposed method achieves good results on CIFAR-LT, ImageNet-LT, and iNaturalist 2018.

**Limitations And Societal Impact:**

The paper aims to address long-tailed recognition. There seems no obvious negative societal impact.

**Main Review:**

The novelty of this paper is limited:

This work is very similar to previous papers [1] and [2]. Concretely, the proposed Bayias loss (Eq.15) is the same as logit adjustment [1] (Eq. 11 in [1]) for long-tailed recognition. [2] analyze long-tailed recognition from the calibration view. Algorithm 1 in this paper is also the same as [2], i.e., adding mixup in Stage-1 and using re-balanced loss in Stage-2. What's more, the whole paper emphasizes calibration. However, the authors just give visualization results (Fig. 1, Fig. 3, and Fig. 5) to illustrate.  Why not show quantitative results (Expected Calibration Error, Brier Score, and Probabilistic sharpness) in experiment part? A lot of theoretical analysis seems meaningless.


The experiment results are not significant:

Balanced Softmax [3] (experiment results please refer to [4]) can get **45.1%** top-1 accuracy on CIAFR-100-LT, $\rho=100$. While the proposed Bayias loss just gets **43.52%** (**both without mixup**).  Balanced Sofemax [3] (experiment results please refer to [4]) **without mixup** can get **69.8%** top-1 accuracy on iNaturalist 2018. While the proposed Bayias loss **with mixup** just gets **69.15%**.

For calibration, the proposed method is **overall lower** than MiSLAS [2] in terms of **both Expected Calibration Error and top-1 recognition accuracy** on all long-tailed benchmarks. It is a reasonable and related fair comparison because both two models add mixup operation.


Other questions:

1) Based on Algorithm 1, I don't think the proposed method is E2E. Since it also contains two stages, which is similar to LDAM+DRW (Table 1: LDAM+DRW E2E  [x])

2) Both UniMix Factor and UniMix Sampler generate samples, which is mainly a $\xi$-Aug sample of the tail composite with $x_i$ from the head. I don't think more balanced $\xi$-Aug samples will lead to better model recognition performance.

3) What is the definition of $f$ in Eq. (6)?


Ref:

[1] Long-tail Learning via Logit Adjustment, ICLR 2021.

[2] Improving Calibration for Long-Tailed Recognition, CVPR 2021.

[3] Balanced Meta-Softmax for Long-Tailed Visual Recognition, NeuIPS 2020.

[4] Disentangling Label Distribution for Long-tailed Visual Recognition, CVPR 2021.

**Time Spent Reviewing:**

10

---

> ### Author Response · Authors · 2021-08-10
> **Author Response to Reviewer 1uZE**
>
> We sincerely appreciate the valuable and professional review by Reviewer 1uZE. Here, we want to rebut some criticisms that are critical to understanding our contribution:
>
> *\# 1. The novelty and contribution of our work.*
> --------------------------------------------
>
> We'd like to verify the difference between ours and previous works[1],[2], the reasons lie in:
>
> a. The proposed Bayias is different with logit adjustment (LA). Instead, Bayias is a modification and generalization of LA. We've made a detailed discussion with various losses in Appendix B.3 (specifically, Appendix Line 135-147 discusses the difference between LA and Bayias).
>
> We need to clarify that Eq.11 in [1] is uniform for any pairwise margin loss. The proposed Bayias-compensated loss (Eq.15) is a specific result that is different from LA. Our original form is available in Appendix B.3 (Eq.B17).
>
> Bayias is a modification and extension of LA. The difference between LA and Bayias are:
>
> - LA contains a scale factor as a hyper-parameter to adjust the margin influence. In contrast, according to our prove, LA misses the $\log (C)$ term, and the scale factor should be identically equal to 1 rather than a hyper-parameter. The authors of LA conduct an ablation study and give the optimal value of the scale factor under different datasets. However, it is a **post-hoc** adjustment procession. In end-to-end training, scale factor equal to 1 is the best case, which is consistent with our analysis.
>
> - LA loss requires balanced distribution of test set and is powerless in other scenarios, while Bayias is capable with imbalanced distributed test sets, i.e., the constant term $\log (C)$ can change to be the test set distribution $\log (\pi')$ to deal with such scenarios.
>
> b. Our method has different motivations compared to MiSLAS. Since [2] are published around the deadline of submission, the motivation of ours and [2] are different. We'd illustrate the difference to expound our novelty.
>
> - Although we all aim at improving calibration, the starting point is completely different. [2] aims to improve the training strategy in 2-stage methods, i.e., it uses Label Aware smooth loss + Shift BN learning + cRT training manner to improve model calibration and accuracy, mainly focusing on the 2nd-stage. However, our 1-stage method originates from the data distribution prior and lies in mathematics proves. As a result, it can also adopt into [2]'s 2-stage methods for fair comparisons.
>
> - We have different algorithm compared to 2-stage MiSLAS. We'd like to clarify the misunderstanding of what 2-stage approaches are: disentangle feature extractor and classifier learning and optimize them separately. The most prominent feature is that these methods (e.g., cRT, MiSLAS) will freeze part of the network or change loss functions to fine-tune in the 2nd-stage. In contrast, our 1-stage method doesn't make such training progress. Instead, our algorithm optimizes the feature extractor and classifier in a single stage, nothing but removes UniMix in the last few epochs as a common trick. As a result, the loss function, network, and data-sampler never change during the whole training process.
>
> [1]. Long-tail Learning via Logit Adjustment, ICLR 2021.
>
> [2]. Improving Calibration for Long-Tailed Recognition, CVPR 2021.
>
> *\# 2. Quantitative comparisons about calibration.*
> --------------------------------
>
> We’ve already made quantitative comparisons of ECE and MCE in Appendix D.2 to show how our methods improve calibration. We sincerely thank the suggestion on additional metrics, and we make the other results w.r.t. **ACE**, **TACE**, **SCE**, and **Brier Score**. Please **refer to Reviewer GN8J**.
>
> *\# 3. Effect of theoretical analysis.*
> --------------------------------
>
> Our theoretical analysis aims to illustrate the limitation of naïve mixup since there is seldom quantitative proof of mixup in LT scenarios before. We also use Bayesian Theory to demonstrate the existing bias (i.e., Bayias) and extend it into any distribution compared with LA. These theoretical analyses enable us to confirm the effectiveness and reliability of our methods.
>
>
> *\# 4. Explains of the experiment results in Balanced Meta-Softmax and MiSLAS.*
> --------------------------------------------------------------------------
>
> a. Comparison with Balanced Meta-Softmax.
>
> First of all, Our method is UniMix+Bayias. Therefore, compare either of them separately with other methods is improper.
>
> We should emphasize the importance of experimental setting, since the results in [4] is under different settings, e.g., different initial learning rate (i.e., **0.1** ours vs. **0.2** in [4]), more train epochs (i.e., **90** epochs ours vs. **200** epochs in [4]) et al. In addition, Balanced Meta-Softmax utilizes meta-learning, which makes the baseline in [3] significantly higher than most ERM baselines in other work and ours. Thus, it's an unfair comparison.
>
> To verify our methods are significant, we reproduce the Balanced Meta-Softmax under the same settings as ours for fair comparisons. The straightforward implementation of Balanced Meta-Softmax is to times instances number with corresponding logits in Softmax operation. However, it encounters coverage problems. Therefore, we refer to the official code implement and find that they convert the instance number to a Softmax logit margin, which is similar to LA loss. The compared results are:
>
> | Dataset / Accuracy | CIFAR-10-LT |           | CIFAR-100-LT |           |
> | ------------------ | :---------: | :-------: | :----------: | :-------: |
> | $\rho$             |     10      |    100    |      10      |    100    |
> | BMSoftmax          |    88.73    |   78.45   |     58.6     |   43.68   |
> | Ours               |  **89.66**  | **82.75** |  **61.25**   | **45.45** |
>
> b. Comparison with MiSLAS.
>
> MiSLAS is released near submission, so we don't report comparisons with MiSLAS in the current version. But we'll add detailed comparisons and discussions in the final version.
>
> MiSLAS is a 2-stage method that integrates Label Aware smooth loss + Shift BN learning + cRT training manner. The baseline cRT far outperforms our ERM baseline, while our method is trained in 1-stage. So it's unfair to compare to ours w.r.t 2-stage vs. 1-stage, cRT vs. ERM, 200+10 epochs vs. 200 epochs.
>
> For fair comparisons, we train our method in 2-stage as MiSLAS does. Thus, our method is on par with MiSLAS performance when fairly compared. Please **refer to Reviewer V6xE** for more detailed discussions and results.
>
> [3]. Balanced Meta-Softmax for Long-Tailed Visual Recognition, NeuIPS 2020.
>
> [4]. Disentangling Label Distribution for Long-tailed Visual Recognition, CVPR 2021.
>
> *\# 5. Answer to other questions.*
> ------------------
>
> Thanks a lot for letting us know your confusion. We'd explain these questions for a comprehensive understanding of our work.
>
> a. Reasons to verify the proposed method is E2E.
>
> 2-stage method is the one that disentangles feature learning and classifier learning [5]. The distinguishing feature is that different loss functions or models are adapted in each stage, e.g., LDAM+DRW will adjust the sampler or reweight loss in the 2nd-stage to fine-tune the model.
>
> However, our method is a 1-stage method because we only remove UniMix in the last few epochs as a trick (we've explained the reason to do this in Line 163-165: mixup methods usually need more epochs for convergence, we follow [6] to remove mixup in the last epochs to ensure the same training epochs for fair comparisons while ensuring accuracy). The loss function and model are never changed during the whole training process.
>
> b. Why more balanced $\xi$-Aug samples will lead to better model recognition performance?
>
> Thanks for raising your concern about the effectiveness of balanced $\xi$-Aug samples. Actually, we've discussed the influence of UniMix Sampler parameter $\tau$ (see Line 231-239) for the same concern. In challenging datasets, we should consider improving feature learning in both the head and tail. $\tau$ should be adjusted to generate head samples and tail samples simultaneously to improve performance.
>
> The motivation of UniMix is to handle the head-head issue of the previous mixup methods. The reason mixup makes sense lies in its improvement of feature learning. However, mixup tends to generate more head-head samples that improve the head regardless of the tail, which leads to its bias and poor generalization of the tail. UniMix aims in favor of tail feature learning, which is the essential reason why UniMix makes sense to achieve better performance.
>
> c. What is the definition of $f$ in Eq.(6)?
>
> $f$ is the **probability density function of the Beta distribution** (see Line 95). $f$ is used to design UniMix factor $\xi^*$. The detailed motivation is discussed in Appendix E (Line 291-298). The key is to improve the robustness and generalization of UniMix.
>
> [5]. Decoupling Representation and Classifier for Long-tailed Recognition, ICLR 2020.
>
> [6]. Bag of Tricks for Long-Tailed Visual Recognition with Deep Convolutional Neural Networks, AAAI 2021.

---

> > ### Comment · Reviewer_1uZE · 2021-08-18
> > **Thanks for your response**
> >
> > Thank you for the detailed response.
> >
> > **1. One-stage/two-stage methods and poor experiment results**
> >
> > I don't think it makes much sense to divide a method into one-stage and two-stage as you said. Because the current two-stage model (LWS, cRT, DRW-LDAM, DRS-LDAM) does not have a lot of extra computation costs (about 5\%). It's not like the two-stage model in other fields (detection) will be much slower (training/inference time) than the single-stage model. Moreover, judging from the experimental results you submitted, the proposed model has no obvious advantage over balanced softmax + mixup (end-to-end training). According to my own experimental results, mixup + balanced softmax can achieve 47.5\~48.5\% accuracy on CIFAR100, $\rho=100$ (3\% higher than your model's). It can achieve 52\~53\% accuracy on ImageNet-LT (4\% higher than your model's). It can achieve 70\~70.5\% accuracy on iNaturalist-2018 (1\% higher than your model's).
> >
> > **2. The novelty: comparing with logit adjustment (LA)**
> >
> > As the authors mentioned, the main difference between LA and the proposed method is the constant term $\log(C)$, which can be changed according to the test set distribution. However, in practice, the class distribution of the test set is difficult to know in advance. Thus, researchers care more about the precision of the average category, for example, MAP for object detection, MIoU for semantic segmentation. There are very few cases that the proposed method can be applied. Therefore, I don't think such a small modification made by the authors can be accepted by the NeurIPS conference.
> >
> > **3. The novelty: comparing with MiSLAS**
> >
> > First of all, the paper's motivation (calibration), some conclusions, and algorithm framework are similarly consistent with MiSLAS. Secondly, the author emphasizes that the proposed method is one-stage and beneficial. However, according to the experimental table mentioned in the author's response to Reviewer V6xE, the author's method (single-stage) hardly surpasses MiSLAS in Accuracy and ECE. Finally, the author avoids making a comparison with MiSLAS. In fact, MiSLAS released the OpenReview version (https://openreview.net/forum?id=Sx-mvOvnmJj) in Sep. 2020. the arXiv version can be viewed on Apr. 1st (https://arxiv.org/abs/2104.00466). While the deadline for NurIPS is May 26th.
> >
> > Based on the above problems, I think it is inappropriate to accept this paper now.

---

> > > ### Author Response · Authors · 2021-08-24
> > > **Author Response to Reviewer 1uZE**
> > >
> > > We sincerely thank Reviewer 1uZE's response and would like to reply to the concerns and doubts.
> > >
> > > *\# 1. Why do previous works divide methods into 1-stage and 2-stage when making comparisons?*
> > > -------------
> > >
> > > We agree that dividing a method into 1-stage and 2-stage is unnecessary when deploying them because 2-stage methods will not introduce too much computation costs.
> > >
> > > However, most previous works will distinguish these approaches because balanced fine-tuning in the 2nd-stage will always achieve further performance gains, e.g., in CIFAR-10-LT-200, LDAM (1-stage) achieves **69.50\%** accuracy, while LDAM+DRW (2-stage) can improve accuracy up to **74.74\%**. Hence, for research purposes, it's necessary to distinguish 1-stage and 2-stage methods for fair comparisons.
> > >
> > > *\# 2. Results of BMS+mixup when fairly compared.*
> > > -------------
> > >
> > > In balanced test set, the numerical value of Bayias is equivalent to LA and BMS mathematically, so **the results of LA, BMS, and Bayias are similar as we reported**. Notice that the baseline reported in [1] is higher than most of previous works (The baseline of CIFAR-100-LT-100 is **45.30\%** in [1] while **38.32\%/38.32\%/38.36\%** in [2,3,4]). We suspect that the performance gap mentioned by Reviewer 1uZE may come from the different ERM baseline.
> > >
> > > To dispel Reviewer 1uZE's doubts, we compare BMS+mixup under different official repos (i.e., from LDAM [2] and from MiSLAS [5]). As shown below, the ERM baseline is different over the two methods. In CIFAR-100-LT-100, the accuracy of ERM in repo 1 is **38.32\%** while it up to **40.41\%** in repo 2. The reproducible difference in the two repos has reached about **2\%**. We therefore advocate a fair comparison with the same baseline. The results show that our proposed method is effective when fairly compared.
> > >
> > > | repos from    |  ERM  |  BMS  | mixup+BMS |  Ours | Ours(2-stage) |
> > > |---------------|:-----:|:-----:|:---------:|:-----:|:-------------:|
> > > | LDAM et al.   | 38.32 | 43.68 |   44.44   | 45.45 |     46.12     |
> > > | MiSLAS et al. | 40.41 | 44.90 |   47.03   | 47.60 |     48.30     |
> > >
> > > *\# 3. Explains of why Bayias make sense and our contribution.*
> > > -------------
> > >
> > > Ignoring the different distribution of train and test sets will exacerbate bias issues and harm classification results (please refer to Reviewer V6xE for the comparisons under different distributed test sets). Such a test prior related term can alleviate the bias by considering distribution difference.
> > >
> > > Apart from Bayias, we theoretically reveal an important head-head pair bias in the existing mixup-based LT method and propose UniMix to tackle it. We also provide a heuristic result that may be insightful for future work.
> > >
> > > *\# 4. The comparison with MiSLAS*
> > > -------------
> > >
> > > As we explained before, although we all aim at improving model calibration, the methodology, conclusions, and algorithm framework are different compared to MiSLAS. We summarize the main difference in the following table for clarity.
> > >
> > >
> > > |  Methods | framework    |   1st-stage  |        |   2nd-stage  |
> > > |----------|----------|:------------:|:------:|:------------:|
> > > |          |          | mixup-method |  loss  |              |
> > > | MiSLAS   | cRT/LWS* |   mixup*     |   CE*  | **LAS/shift BN** |
> > > | Ours     | -        |    **UniMix**    | **Bayias** |       -      |
> > >
> > > (*) indicates off-the-shelf methods that MiSLAS adopts, (**bold type**) represents the proposed methods, respectively.
> > >
> > > The detailed discussions and comparisons will contain in the final version paper.
> > >
> > > [1] Balanced Meta-Softmax for Long-Tailed Visual Recognition, NeurIPS 2020.
> > >
> > > [2] Learning Imbalanced Datasets with Label-Distribution-Aware Margin Loss, NeurIPS 2019.
> > >
> > > [3] BBN: Bilateral-Branch Network with Cumulative Learning for Long-Tailed Visual Recognition, CVPR 2020.
> > >
> > > [4] Long-Tail Learning via Logit Adjustment, ICLR 2021.
> > >
> > > [5] Improving Calibration for Long-Tailed Recognition, CVPR 2021.

---

### Official Review · Reviewer_GN8J · 2021-07-14

**Rating:** 6
**Confidence:** 2

**Summary:**

The authors proposed a new method based on two approaches, UniMix and Bayias, to sample input-output pairs from unbalanced datasets and use them in Mixup. The authors report experiments on CIFAR-10-LT, CIFAR-100-LT, ImageNet-LT, and iNaturalist 2018, showing superior performance in terms of accuracy and calibration scores over a variety of baselines.

While I could get the main idea behind the proposed method, which seems appealing, I had problems understanding all the details, mainly because of a substantial lack of clarity. Sentences are often disconnected which makes it difficult to understand what the authors mean.  My score is borderline but I am open to increase it if the authors can provide a solid rebuttal and substantially improve the paper in terms of clarity.

**Limitations And Societal Impact:**

Not reported. This work has minimal societal impact.

**Main Review:**

Strengths
---------

- The method seems to follow a different rebalancing approach compared to other work (e.g. Remix [1]).
- The empirical results show a substantial improvement in terms of accuracy over other baselines.
- Comparison in terms of ECE scores confirm that the method improves calibration.

Weaknesses
----------

1) The paper lacks in terms of clarity and it is often difficult to understand what the authors mean. The paper needs to be substantially improved from this point of view.

2) One of the main claims of the authors is that the proposed method improves the model calibration w.r.t. Mixup on unbalanced datasets. The plots reported by the authors seem to suggest that this is true, but those plots show the accuracy VS confidence tradeoff, that can be misleading. In Appendix D.2 the authors report quantitative measurements in terms of ECE and MCE. It has been showed that ECE scores suffer of several shortcomings [3]. I think that it would be useful to provide a quantitative comparison also in terms of other metrics like ACE and TACE scores [2], which are more robust than ECE.

3) It is unclear why best performances and calibration are obtained by joining UniMix+Bayias. The authors provided an ablation study in Table 3 but there is not a proper discussion to explain these results.

4) The authros claim (lines 42-48) that one of the main contributions is the introduction of the $\\xi\\text{-Aug}$ notation for Mixup (Section 2). I am struggling to understand where is the novelty, since it seems to me that this is a fairly common notation used to describe Mixup. Could the authors expand on this point?

5) All the experimental results do not report the standard deviation over accuracy. Are the reported results just on a single seed? Please provide additional details regarding this point.

6) There is not a proper discussion of the shortcomings of this method. It seems to me that joining UniMix+Bayias has a higher computational overhead compared to other approaches (e.g. Remix [1]). I would like to see a discussion about this point.


References
----------

[1] Chou, H. P., Chang, S. C., Pan, J. Y., Wei, W., & Juan, D. C. (2020, August). Remix: Rebalanced mixup. In European Conference on Computer Vision (pp. 95-110). Springer, Cham.

[2] Nixon, J., Dusenberry, M. W., Zhang, L., Jerfel, G., & Tran, D. (2019, June). Measuring Calibration in Deep Learning. In CVPR Workshops (Vol. 2, No. 7).

[3] Ashukha, A., Lyzhov, A., Molchanov, D., & Vetrov, D. (2020). Pitfalls of in-domain uncertainty estimation and ensembling in deep learning. arXiv preprint arXiv:2002.06470.

**Time Spent Reviewing:**

5

---

> ### Author Response · Authors · 2021-08-10
> **Author Response to Reviewer GN8J**
>
> We sincerely appreciate the valuable suggestion given by Reviewer GN8J. We've asked established experts to polish our paper and will improve the clarity in the final version. Our responses to the reviewer’s questions and concerns are as follows:
>
> *\# 1. To clarify our motivation and contributions.*
> ----------------------------------------------------
>
> We'd elaborate the motivation and contributions of this paper, i.e., we propose UniMix and Bayias theoretically towards better calibration from prior perspective in LT scenarios.
>
> Specifically, we first illustrate the limitations of mixup in LT scenarios and theoretically prove the proposed UniMix enables a more balanced distribution in favor of tail feature learning.
>
> Then, we demonstrate the existence of bias between imbalanced dataset (noteworthy: not only long-tailed distribution, but for any scenarios that the different distributed train and test set). And we propose the Bayias-compensated CE loss to relieve the bias influence from the head towards the tail.
>
>
> *\# 2. Explains on the acc vs. conf plots and additional quantitative comparisons.*
> -----------------------------------------------------------------------------------
>
> Calibration means that the predicted confidence indicates actual accuracy likelihood. Hence, while the real confidence gets closed to its prediction accuracy, solid calibration is ensured. From another perspective, a well-calibrated model will neither be overconfident nor underconfident of each class. Previous work [1] also adopts acc vs. conf plots as we do. We are very sorry that our expression has confused you. We hope the above explanation can help you better understand what we mean.
>
> We also sincerely thank the reviewer for inspiring suggestions. We make additional quantitative calibration comparisons in terms of **ACE, TACE (threshold=1e-3), SCE, and Brier Score** below. The proposed methods still outperform other existing techniques in these above metrics.
>
> | Dataset                           |          |          | CIFAR-10-LT-100 |                 |          | CIFAR-100-LT-100 |           |                 |
> | --------------------------------- | :------: | :------: | :-------------: | :-------------: | :------: | :--------------: | :-------: | :-------------: |
> | Calibration Metric ($\times$ 100) | **ACE**  | **TACE** |     **SCE**     | **Brier Score** | **ACE**  |     **TACE**     |  **SCE**  | **Brier Score** |
> | ERM                               |   5.42   |   4.66   |      5.46       |      53.61      |  0.831   |      0.709       |   0.952   |      97.05      |
> | mixup                             |   4.87   |   4.20   |      4.92       |      49.09      |  0.751   |      0.683       |   0.849   |      89.26      |
> | Remix                             |   3.49   |   3.32   |      3.77       |      44.84      |   0.740   |      0.675       |   0.846   |      89.33      |
> | LDAM+DRW                          |   4.14   |   2.84   |      4.43       |      44.76      |  0.801   |      0.671       |   1.093   |      107.4      |
> | UniMix (ours)                     |   3.48   |   3.27   |      3.57       |      38.98      |  0.505   |      0.555       |   0.687   |      80.70      |
> | Bayias (ours)                     |   2.45   |   2.40   |      2.69       |      34.30      |   0.460   |      0.518       |   0.631   |      78.78      |
> | UniMix+Bayias (ours)              | **2.31** | **2.17** |    **2.43**     |    **32.84**    | **0.450** |    **0.517**     | **0.623** |    **78.24**    |
>
> [1] On Mixup Training: Improved Calibration and Predictive Uncertainty for Deep Neural Networks, NeurIPS2019
>
> *\# 3. The reasons why best performances and calibration are obtained by joining UniMix+Bayias.*
> ------------------------------------------------------------------------------------------------
>
> When we combine different mixup strategies and loss modification methods to obtain higher accuracy, some combinations (e.g., Remix+LDAM) fail to achieve satisfactory performance gains. We further analyze the calibration of each method and find that the well-calibrated methods seem synergistic, and their combination may further improve the top-1 accuracy. Based on such observation, we design the UniMix to improve the calibration of mixup, and propose Bayias-compensated CE loss to correct the bias caused by the different distributed train and test sets. Inspired by [2], we find such loss contributes to a better-calibrated model. As a result, UniMix+Bayias outperforms other methods.
>
> Here we'd provide the explanations: the challenges of LT are two-fold. One is that the paucity of tail data leads to insufficient feature learning, and the other is that the head categories suppress the tail ones. Previous work [1] proves that mixup is helpful for feature learning. However, in LT scenarios, mixup improves the model feature learning mainly in the head and fails to deal with the bias. The results in Fig.5 D4-D7 also prove that. Some works [3,4] try to handle it by 2-stage training. However, we attempt to solve such problems in an end to end way by proposing UniMix to improve the frequency of tail samples in mixup manner.
>
> The approaches to tackle LT problem are essentially to impose different constraints on different classes. Our UniMix and Bayias loss both impose constraints for each class according to its label priors, which may be consistent constraints from this perspective.
>
> [2] Long-Tail Learning via Logit Adjustment, ICLR2021
>
> [3] Decoupling Representation and Classifier for Long-tailed Recognition, ICLR2020
>
> [4] Improving Calibration for Long-Tailed Recognition, CVPR2021
>
> *\# 4. The reasons why $\xi$-Aug is one of the main contributions in this work.*
> --------------------------------------------------------------------------------------------------------------------------------------------------------------------------------------------------------------------------------------------------------------------------------------------------------------------
>
> Thanks for letting us know your confusion. We'd like to explain the definition to avoid misunderstanding. The definition of $\xi$-Aug is **not a notation for mixup**. Instead, it's a novel definition to represent and measure the **mixed samples** (instance + label) (see Line 66-68) given by mixup.
>
> Although it is intuitive to conclude that mixup generates more head-head samples, such a conclusion lacks quantitative proof and analysis. The novelty is: our assumption $\xi$-Aug enables us to make mathematical proves for the distribution of virtual mixed samples. Based on it, we can analyze the mixup's limitation in LT scenarios and hence deduce our UniMix strategy. Also, we demonstrate the reliability of $\xi$-Aug in Experiments 1 (see Line 167-182). In conclusion, since we are the first to propose the concept of $\xi$-Aug, we would like to highlight $\xi$-Aug as our significant contribution. Indeed, the above problem is worth further study.
>
> *\# 5. All the experimental results do not report the standard deviation over accuracy. Are the reported results just on a single seed?*
> ----------------------------------------------------------------------------------------------------------------------------------------
>
> We report experiments to keep consistent with most previous works (e.g., [2,3,4,5,6]), which don't report standard deviation as well. In our experiments, the reported results are average values on 5-repeat experiments rather than on a single seed.
>
> [5] Learning Imbalanced Datasets with Label-Distribution-Aware Margin Loss, NeurIPS2019.
>
> [6] Class-Balanced Loss Based on Effective Number of Samples, CVPR2019.
>
> *\# 6. Shortcomings and computational overhead discussion.*
> -----------------------------------------------------------
>
> We illustrate the shortcoming and future work of our method in Line 290, i.e., remains theoretical proves to account for our empirical observation in ablation studies, w.r.t. calibration-ensured methods combined tend to improve accuracy while poor-calibrated methods may be counter-productive when combined with others.
>
> We'd like to verify our method won't introduce additional computational overhead. The reasons are:
>
> Compared with Remix mentioned, our method contains a modified mixup strategy and loss, i.e., UniMix and Bayias, whereas Remix is just a single mixup strategy. As a result, **we compare UniMix with mixups and Bayias-compensated CE with standard CE**.
>
> - a. Naïve mixup mixes two inputs and combines the corresponding labels in proportion into a non-one-hot vector. When computing loss, it calculates the loss of the new mixed input on both label y1 and y2. Then the final loss is the proportional addition of the two losses. Hence, for an virtual sample, loss will be calculated twice. Remix proposes a threshold to decide the pseudo label to be mixed or totally belong to one of the input pairs, so the loss is computed once or twice depend on label. Hence, Remix has slightly less calculation. UniMix improves the sampling strategy and mixing strategy, and it computes loss twice as mixup does. So in detailed comparison, UniMix has about the same computational overhead as mixup, slightly more than Remix (i.e., additional one loss calculation at most).
>
> - b. Compared with standard CE loss, Bayias modifies CE loss by introducing prior related constant, which is known before training (see Appendix Eq.B17). As a result, it won't have additional computational overhead. Additionally, the novelty of Bayias lies in: previous methods must default the test set to be balanced. However, the inference case is always not balanced distributed in the real world (corresponding to $\log (C)$ term in Bayias). Thanks to existing approaches to estimate the distribution of the real world data, Bayias is still able to handle the bias issue while other methods (e.g., logit adjustment) fail when the test set is not balanced distributed (see detailed discussion in Appendix Line 299-310).

---

> > ### Comment · Reviewer_GN8J · 2021-08-17
> > **Comment to authors' response**
> >
> > Thank you for the detailed answer. I have read the response of the authors and the comments of the other reviewers. Most of my concerns have been addressed in the rebuttal. However, there is a substantial work to be done in terms of clarity, I invite the authors to carefully review the manuscript for the final version. Given those limitation and my confidence level I will only increase my score to a 6.

---

### Official Review · Reviewer_Xiu3 · 2021-07-16

**Rating:** 7
**Confidence:** 3

**Summary:**

This paper addressed the problem of long tail recognition using a novel mixup technique. The proposed mixup technique (Unimix) considered the class prior and helps to create a more balanced virtual dataset. The author provide theoretical analysis on why Unimix is a better mixup technique. Extensive experiment also demonstrates the performance gain of using Unimix.

**Limitations And Societal Impact:**

The authors have adequately addressed the limitations and potential negative societal impact of their work.

**Main Review:**

Strength
a.	The paper is well written and all the figures are well illustrated.
b.	The theoretical analysis is interesting and provide strong support to the proposed method.
c.	The experiment is thorough and cover different aspects of the ablation studies.
d.	The performance is promising

Minor weakness
a.	Although the author somehow shows the performance on head/medium/tail classes using the confusion matrix, it will be nice to provide the number and table in the final version.


**Time Spent Reviewing:**

2

---

> ### Author Response · Authors · 2021-08-10
> **Author Response to Reviewer Xiu3**
>
> *\# 1. Although the author somehow shows the performance on head/medium/tail classes using the confusion matrix, it will be nice to provide the number and table in the final version.*
> -------------
>
> We sincerely thank Reviewer Xiu3’s support and constructive comments. Apart from the novel UniMix, this paper also introduces a noticeable bias between the train dataset and real test dataset and compensates Bayias on standard cross-entropy loss for better calibration. We've provided the detailed number of CIFAR-10-LT confusion matrices in Fig.D4-D5. Additional tables and values of other results will be available in the final version supplementary material. Thanks a lot for your suggestion.

---

### Decision · Program_Chairs · 2021-09-27

**Decision:**

Accept (Poster)

**Comment:**

This paper addresses the problem of long-tailed visual recognition. They proposed two complementary methods to address the imbalance problem: (i) a data augmentation scheme based on the Mixup and (ii) an approach to compensate for bias in class prior. The final proposed pipeline is 1 stage, as opposed to 2-stage which has featured prominently in recent work. Results are presented across several different image classification datasets, showing superior performance compared to existing work in most cases.

During the review process there was a lot of discussion regarding the relative performance compared to the recent CVPR’21 paper, “Improving Calibration for Long-Tailed Recognition” AKA MiSLAS. This paper was published after the NeurIPS submission deadline, and thus the relationship between the two works did not factor into the final decision. However, the authors are strongly encouraged to cite it, and include the new 2-stage comparison to MiSLAS from the response for completeness. In addition, they should more clearly clarify the relationship to the logit adjustment work from ICLR’21.

The reviewers noted that while the technical contributions were not overly significant, the main insight of the paper (that naive mixup introduces “head to head” class bias, i.e. a bias towards head majority pseudo data) is well characterized and will be of interest to the long-tail community. There were also concerns about the clarity of the exposition and some missing discussion. Many of these issues were cleared up in the discussion, but the authors are strongly encouraged to address these issues with the writing for the final version. This is very important as it will hopefully significantly improve the clarity of the paper. The paper should also be proof-read for grammar (e.g. Line 129).

In the end, three of the reviewers were broadly supportive of the paper, but one recommended rejection (based mostly on the relationship to MiSLAS - see above). This AC, agrees with the consensus of the reviewers and supports the paper being accepted.